

# Investigating the role of dust in ice nucleation within clouds and further effects on regional weather system over East Asia

# Part I: model development and validation

Lin Su[1], and Jimmy C.H. Fung[2, 3]

[1] School of Science, Hong Kong University of Science and Technology, Hong Kong, China

[2] Division of Environment, Hong Kong University of Science and Technology, Hong Kong, China

[3] Department of Mathematics, Hong Kong University of Science and Technology, Hong Kong, China

*Correspondence to: Lin Su (lsu@connect.ust.hk)*

**Keywords:** dust; ice nucleation; microphysics scheme implementation; numerical modeling

**Highlights:**

An aerosol model has been coupled with a microphysics scheme for evaluating the role of dust particles in atmospheric ice nucleation.

The effect of dust on atmospheric ice water content over East Asia during a dust-intensive period is simulated.

The simulation of atmospheric ice water content during dust events is substantially improved upon the effect of dust being considered.





**Abstract.** The GOCART–Thompson microphysics scheme, which couples the Goddard Chemistry Aerosol Radiation
and Transport (GOCART) model and aerosol-aware Thompson microphysics scheme, has been implemented in the
Weather Research and Forecast model coupled with Chemistry (WRF-Chem), to quantify and evaluate the effect of
dust on the ice nucleation process in the atmosphere by serving as ice nuclei. The performance of the GOCART-
Thompson microphysics scheme in simulating the effect of dust in atmospheric ice nucleation is then evaluated over
East Asia during spring in 2012, a typical dust-intensive season. Based upon the dust emission reasonably reproduced
by WRF-Chem, the effect of dust on atmospheric cloud ice water content is well reproduced. With abundant dust
particles serving as ice nuclei, the simulated ice water mixing ratio and ice crystal number concentration increases by
one order of magnitude over the dust source region and downwind areas during the investigated period. The
comparison with ice water path from satellite observations demonstrated that the simulation of cloud ice profile is
substantially improved by applying the GOCART–Thompson microphysics scheme in the simulations. Additional
sensitivity experiments are carried out to optimize the parameters in the ice nucleation parameterization in the
GOCART–Thompson microphysics scheme, and the results suggest that the calibration factor in the ice nucleation
scheme should be set to 3 or 4. Lowering the threshold relative humidity with respect to ice to 100% for the ice
nucleation parameterization leads to further improvement in cloud ice simulation.



## 1 Introduction

As one of the largest natural aerosol sources, dust aerosol contributes considerably to the global aerosol burden (Textor et al., 2006). The Intergovernmental Panel on Climate Change (IPCC) has recognized dust as a major component of atmospheric aerosols, which are an "essential climate variable." East Asia is a major contributor to the Earth's dust emission. It has been reported in previous studies that East Asian dust contributes 25–50% of global emission, depending on the climate of the particular year (Ginoux et al., 2001).

Dust in the atmospheric can alter the Earth's radiation budget through certain ways. By reflecting, absorbing and scattering the incoming solar radiation, dust can cause a warming effect within the atmosphere, and a cooling effect at the surface layer (Lacis, 1995). Dust within clouds can absorb short-wave and long-wave radiation and heat up the surrounding environment, causing a shrinking of cloud, and a lower cloud albedo (Perlwitz and Miller, 2010;Hansen et al., 1997). Moreover, dust particles are recognized as effective ice nuclei, and play an important role in the ice nucleation process in the atmosphere, directly affecting the dynamics in ice and mixed-phase clouds, such as the formation and development of clouds and precipitation (Koehler et al., 2010;Twohy et al., 2009).

To date, Many studies have been conducted to evaluate the direct radiative effect of dust aerosol using radiation schemes implemented in numerical models all over the world (Mallet et al., 2009;Nabat et al., 2015a;Ge et al., 2010;Hartmann et al., 2013;Huang et al., 2009;Bi et al., 2013;Liu et al., 2011a;Liu et al., 2011b;Huang, 2017). Recently, semi-direct effect of dust has been investigated in a few studies over different regions by applying various global and regional models (Tesfaye et al., 2015;Nabat et al., 2015b;Seigel et al., 2013). Unfortunately, due to the poor understanding on the dust-cloud-interaction in microphysics processes, quantifying the microphysical effect of dust remains as a difficult problem. Various ice nucleation parameterizations have been implemented into global models to estimate the importance of dust in atmospheric ice nucleation (Lohmann and Diehl, 2006;Karydis et al., 2011;Hoose et al., 2008;Zhang et al., 2014). However, most regional models are not capable of estimating the indirect effect of dust, and vary rare work has been done to assess the indirect effects of dust on the weather system, especially over East Asia, which is a major contributor to the global dust emission. Currently, only a few microphysics schemes considering aerosol-cloud-interaction are implemented in regional, and in most of these microphysics schemes, only the cloud condensation nuclei served by aerosols are considered (Perlwitz and Miller, 2010;Solomos et al., 2011;Miller et al., 2004), with the ice nuclei not treated, or represented by a prescribed ice nuclei distribution (Chapman et al.,



2009;Baró et al., 2015), and the number of predicted ice crystals is a function of temperature or ice saturation. In
reality, however, the number of ice crystals that can form in the atmosphere is highly dependent on the number of
particles that can act as ice nuclei, and dust is the most abundant aerosols that can effectively serve as ice nuclei and
affect the formation and development of mixed-phase and ice clouds in the atmosphere. This effect should not be
neglected in numerical models, especially in the simulations over arid regions during strong wind events (DeMott et
al., 2003;Koehler et al., 2010;DeMott et al., 2015;Lohmann and Diehl, 2006;Atkinson et al., 2013).
The GOCART aerosol model, which has been implemented in WRF-Chem, is one of the most widely-used aerosol
models for global and regional dust simulation (Chen et al., 2014;Ashrafi et al., 2017;Chiao et al., 2016;Rizza et al.,
2017;Flaounas et al., 2017;Kumar et al., 2014). It is coupled with various radiation schemes for evaluating the radiative
effects induced by dust in WRF-Chem. However, it is not linked with any microphysics scheme, therefore, the indirect
effect of dust cannot be calculated in the model.
Since 2014, the aerosol-aware Thompson microphysics scheme, which takes into account the aerosols serving as ice
nuclei, has been implemented into WRF, enabling the model to explicitly predict the droplet number concentration
for cloud droplets through the number concentrations of cloud condensation nuclei and ice nuclei (Thompson and
Eidhammer, 2014). Therefore, the aerosol-aware Thompson scheme is an ideal microphysics scheme for evaluating
the effect of dust in atmospheric ice nucleation processes. However, this scheme is not coupled with any aerosol model
in WRF-Chem. When the aerosol-aware Thompson microphysics scheme is activated, the model reads in pre-given
monthly-averaged climatological aerosol data derived from the output of other global climate model, which introduces
large errors into the estimation of the effects of dust in microphysical processes, especially in real-time simulations.
In light of the above, we aimed to fully couple the aerosol-aware Thompson microphysics scheme with an aerosol
model in the WRF-Chem modeling system in this study, enabling the model to simulate the effect of dust aerosol in
ice nucleation processes during online simulations, for investigating the role that East Asian dust plays in the ice
nucleation process in the atmosphere.
The remainder of the manuscript is presented as follows. Section 2 provides a description of the model including the
implementation work for coupling the aerosol-aware Thompson microphysics scheme and the GOCART aerosol
model in WRF-Chem, followed by the model configurations for numerical simulations in section 3. Section 4 presents



the observational data used to validate the performance of the newly–implemented GOCART-Thompson
microphysics scheme. Section 5 is the results and discussion, followed by the conclusions section 6.

## 92 2 Model description

WRF-Chem is an online-coupled regional modeling system, which means that it can simultaneously simulate the
meteorological field, the chemical field, and the interaction in between (Grell et al., 2013). The chemical model
contains several gas- and aerosol-phase chemical schemes. In this study, we focus on the GOCART model, a simple
aerosol model that will be used for dust simulation.

## 98 2.1 GOCART aerosol model

GOCART is an aerosol model for simulating major tropospheric aerosol components, such as sulfate, dust, black
carbon, organic carbon, and sea-salt aerosols (Ginoux et al., 2001;Chin et al., 2000). It has been implemented into
WRF-Chem as a bulk aerosol scheme. GOCART is a simple aerosol scheme that can predict the mass of aerosol
components, but does not account for complex chemical reactions. Therefore, it is numerically efficient in simulating
aerosol transport, and thus applicable to cases without many chemical processes, especially dust events.
Shao's dust emission scheme is one of the three dust emission schemes in the GOCART aerosol model (Kang et al.,
2011;Shao, 2004, 2001;Shao et al., 2011), and has been demonstrated to exhibit superior performance in reproducing
the dust cycle over East Asia compared to other emission schemes (Su and Fung, 2015). The Shao's emission scheme
was updated in WRF-Chem since version 3.8 released in 2016, to produce five size bins for dust emission, with
diameters of < 2 μm, 2–3.6 μm, 3.6–6.0 μm, 6.0–12.0 μm, and 12.0–20.0 μm, and mean effective radii of 0.73 μm,
1.4 μm, 2.4 μm, 4.5 μm, and 8.0 μm.

## 111 2.2 Aerosol-aware Thompson microphysics scheme

The Thompson scheme is a bulk two-moment microphysics scheme that considers the mixing ratios and number
concentrations for five water species: cloud water, cloud ice, rain, snow and a hybrid graupel/hail category. The





aerosol-aware version of the Thompson scheme incorporates the activation of aerosols serving as cloud condensation
nuclei and ice nuclei, and therefore it explicitly predicts the droplet number concentration of cloud water as well as
the number concentrations of cloud condensation nuclei and ice nuclei. Hygroscopic aerosols that serve as cloud
condensation nuclei are referred to as water-friendly aerosols, and those non-hygroscopic ice-nucleating aerosols are
referred to as ice-friendly aerosols. The cloud droplets nucleate from explicit aerosol number concentration using a
look-up table for the activated fraction as determined by the predicted temperature, vertical velocity, number of
available aerosols, and pre-determined values of the hygroscopicity parameter and aerosol mean radius, while the
nucleation of ice crystals by dust particles follows the parameterization of DeMott et al. (DeMott et al., 2010) to
account for condensation and immersion freezing, and the parameterization of Phillips et al. (Phillips et al., 2008) to
account for deposition nucleation. Freezing of super-cooled water droplet is determined following the Bigg's
parameterization (Bigg, 1953), but with a temperature adjustment of a few degrees depending on dust concentration
(Thompson and Eidhammer, 2014). In the current version of WRF-Chem, the number concentrations of both water-
friendly aerosols and ice-friendly aerosols are pre-given in the initialization of the simulations, and are derived from
the climatological data produced by global model simulations in which particles and their precursors are emitted by
natural and anthropogenic sources and explicitly modeled with multiple size bins for multiple species of aerosols by
the GOCART model. In the consequent simulations, a fake aerosol emission is implemented by giving a variable
lower boundary condition based on the initial near-surface aerosol concentration and a simple mean surface wind for
calculating a constant aerosol flux at the lowest level in the model. The number concentrations of both water-friendly
aerosols and ice-friendly aerosols are then updated at every time step by summing up the fake aerosol emission fluxes
and tendencies induced by aerosol-cloud-interaction. The limitation of the current aerosol-aware Thompson scheme
is that the aerosol profile generated from a fake emission can hardly represent the realistic aerosol level, leading to
great errors in quantifying the indirect effects of aerosols.

**2.3 Implementation of GOCART-Thompson microphysics scheme**
To investigate the real-time indirect effects of dust aerosol over East Asia, modifications have been made to couple
the GOCART model with the aerosol-aware Thompson microphysics scheme. The modifications were based on WRF-





Chem version 3.8.1, and consisted of three parts, modification of the GOCART aerosol scheme, modification of the
aerosol-aware Thompson microphysics scheme, and the introduction of a new wet removal scheme.

**2.3.1 Upgraded GOCART aerosol model**
Currently, the GOCART aerosol model generates only the mass concentration for aerosols but no number
concentrations. However, the number concentration of aerosols are required for a microphysics scheme to evaluating
the indirect effects of aerosols. Therefore, modification was needed to provide information about the number
concentrations of aerosols from the mass concentration produced in GOCART aerosol model.
The aerosol mass concentration was converted into number concentration using the aerosol density and effective radius
for each size bin. Assuming that dust particles are spherical, the mass per dust particle ($m_p$, μg/#) for a size bin can
be approximated through the mean effective radius and density for that size bin.
$$m_p = \rho_{dust} \times \frac{4}{3} \times \pi r_{dust}^3 \qquad (1)$$
where $\rho_{dust}$ and $r_{dust}$ are the dust density and mean effective radius, respectively, for a particular size bin.
The aerosol number concentration $N$ (#/kg) for size bin $n$ at a grid point $(i, j, k)$ is then calculated by the following
equation:
$$N(i, j, k, n) = C(i, j, k, n) / m_p \qquad (2)$$
where $C(i, j, k, n)$ is the dust mass mixing ratio $(\mu g/kg)$ for size bin n at grid point $(i, j, k)$. Summing up the aerosol
number concentrations through all of the size bins gives a total dust number concentration, which will be passed into
the Thompson microphysics scheme. Note that all of the dust particles are treated as ice-friendly aerosols in this study,
and represented by a newly-introduction variable, ice–friendly aerosol produced by GOCART aerosol model (*GNIFA*).
$$GNIFA(i, j, k) = \sum_{i=1}^{n} N(i, j, k, n) \qquad (3)$$

**2.3.2 Ice nucleation parameterization**





In the aerosol-aware Thompson scheme, the condensation and immersion freezing above water saturation in the ice
nucleation process is determined following the DeMott's parameterization proposed in 2010 (hereafter DeMott2010
scheme) based on combined data from field experiments at a variety of locations over 14 years (DeMott et al., 2010),
to account for condensation and immersion freezing. In the DeMott2010 scheme, the relationship between ice nuclei
number concentration and ice crystal number concentration is as follows:
$$n_{ice,T_k} = a(273.16 - T_k)^b n_{aero}^{(c(273.16-T_k)+d)} \qquad (4)$$

where $n_{ice,T_k}$ is the ice crystal number concentration (std L$^{-1}$) at temperature of $T_k$; $n_{aero}$ is the number concentration
of ice-friendly aerosols (std cm$^{-3}$), and a, b, c, and d are constant coefficients equal to 5.94×10$^{-5}$, 3.33, 2.64×10$^{-2}$, and
3.33×10$^{-3}$, respectively. The parameterization was tested with various temperatures and number concentration of ice-
friendly aerosols, yielding a good performance in reproducing ice crystal number concentration under conditions of
relatively low mixing ratio of water vapor or low concentration of ice crystals compared with field–experimental data.
The relationship between simulated ice nuclei number concentration and ice crystal number concentration is basically
linear for concentrations of both of under 1,000 std cm$^{-3}$ (DeMott et al., 2010).
The above parameterization was further developed in 2015 (hereafter the DeMott2015 scheme) for conditions of
higher mixing ratio of water vapor or higher concentrations of ice crystals based on the latest data from field and
laboratory experiments. According to the latest observational data, ice crystal number concentration increases
exponentially with ice-friendly aerosol number concentration, and existing aerosols with relatively low concentrations
(less than 1,000 std cm$^{-3}$) can produce a large number of ice crystals (more than 100,000 std L$^{-1}$). The updated
relationship between ice nuclei number concentration and ice crystal number concentration in the Demott2015
parameterization scheme is as follows.
$$n_{ice,T_k} = c_f n_{aero}^{\alpha(273.16-T_k)+\beta} \exp(\gamma(273.16 - T_k) + \delta) \qquad (5)$$

where $\alpha, \beta, \gamma$, and $\delta$ are constant coefficients equal to 0, 1.25, 0.46, and -11.6, respectively. The calibration factor $c_f$
ranges from 1 to 6, and is recommended to be 3. The updated parameterization shows a good performance in
reproducing ice crystal with relatively high concentration in ice nucleation involving ice-friendly aerosols under
varying temperature and aerosol number concentration.





The number concentrations of ice crystals that produced by the DeMott2015 scheme is much higher than that produced
by the DeMott2010 scheme when applied to the same value of $n_{aero}$, and the difference grows larger with decreasing
temperature and increasing number concentration of ice-friendly aerosols (DeMott et al., 2015). As the DeMott2015
scheme has been examined using more comprehensive field– and laboratory–experimental data, we apply the
DeMott2015 ice nucleation scheme in the GOCART–Thompson microphysics scheme, instead of the DeMott2010
scheme in the default aerosol-aware Thompson microphysics scheme to simulate the ice nucleation involving dust.
Originally, the calibration factor $c_f$ is set to be 3; the threshold temperature is set to be –20 °C. The ice nucleation
process is triggered once the relative humidity with respect to ice (RH$_i$) exceeds 105%. Furthermore, when the relative
humidity with respect to water (RH$_w$) is above 98.5%, it is counted as condensation and immersion freezing, and
calculated by DeMott2015 scheme; when RH$_w$ is below 98.5%, it is treated as deposition nucleation, and determined
by the Phillips parameterization (Phillips et al., 2008).

### 200    2.3.3 GOCART-Thompson microphysics scheme

Firstly, the initialization module for the aerosols in the aerosol-aware Thompson scheme was modified. The
initialization module used to read in pre-given climatological aerosol data at the first time step of the simulation, which
provided an annual mean of the global distribution of the number concentrations of the water-friendly and ice-friendly
aerosols in the aerosol-aware Thompson microphysics scheme. In the GOCART–Thompson microphysics scheme,
the initialization module was removed, instead, the scheme was modified to read in the bulk number concentration of
aerosols produced by the GOCART aerosol model updated at every time step.
Secondly, the bulk number concentration of ice-friendly aerosols read in from the GOCART aerosol model is passed
into the GOCART–Thompson microphysics scheme for the calculation of the number concentration of ice nucleating
particles.
Based on the modification above, the deposition rate of the ice-friendly aerosols at grid point *(i, j, k)*, which is the
tendency of ice-friendly aerosols in terms of number concentration due to the ice nucleation process at this grid point,
is calculated.





After the microphysical processes are finished for a particular time step, the tendency term ($ten_{dust}$, #/kg/s) for the
bulk aerosol number concentration produced by the microphysics scheme is then passed into a wet scavenging scheme
to calculate the loss of aerosol mass due to the ice nucleation process, as well as the collision-collection by
precipitations within clouds. Finally, the aerosol mass field can be updated.

**2.3.4 In-cloud wet scavenging**
As no in-cloud scavenging is considered for dust aerosol in WRF-Chem, a new wet scavenging process was then
introduced into WRF-Chem to calculate the loss of aerosol mass due to the microphysical processes within clouds
using the tendency of aerosol number concentration produced by the microphysics scheme. Assuming that the
collection of dust particles is proportional to the number concentration, the fraction of dust particle for each size bin
can be calculated in the GOCART aerosol model:
$$\varphi(i, j, k, n) = \frac{N(i,j,k,n)}{GNIFA(i,j,k)} \qquad (6)$$

The tendency of ice-friendly aerosol is then distributed into each size bin and the loss of dust mass due to the
microphysical processes for a particular size bin *n* is calculated by the following equation:
$$wetscav(i, j, k, n) = ten_{dust}(i, j, k) \times \varphi(i, j, k, n) \times m_p \times dt \qquad (7)$$

where *dt* is the time step for the simulation.
The mass mixing ratio for dust aerosol in a particular size bin *n* is then updated for the following simulation at the
next time step:
$$C(i, j, k, n) = C(i, j, k, n) - wetscav(i, j, k, n) \qquad (8)$$

Apart from the in-cloud scavenging, the below-cloud wet removal is calculated by the default wet deposition scheme
in the GOCART aerosol model, in which the wet removal of dust is proportional to concentration.

**3 Model configurations**



A numerical experiment was conducted to examine the performance of the newly-implemented GOCART–Thompson
microphysics scheme in simulating the ice nucleation process induced by dust in the atmosphere. According to the
observations, the dust events in 2012 were concentrated in mid-March to late-April, and the satellite observations from
mid-March to the end of April were available for model validation; therefore, the simulation period for this numerical
test was from March 9 to April 30, 2012, with the first eight days as "spin-up" time. Only the results from March 17
to April 30, 2012 were used for further analysis. The final reanalysis data provided by the United States National
Center of Environmental Prediction with a horizontal resolution of one degree was used for generating the initial and
boundary conditions for the meteorological fields, and the simulations were re-initialized every four days, with the
aerosol field being re-cycled, which means that the output of the aerosol field from the previous four-day run was used
as the initial aerosol state for the subsequent four-day run.
Two nested domains were used for the simulation, as shown in Figure 1. The outer domain (domain 1) is in a horizontal
resolution of 27 km, and covers the entire East Asia region. The inner domain (domain 2) is in a horizontal resolution
of 9 km, and covers the entire central to East China. Both domains have 40 vertical layers, with the top layer at 50
hPa. The locations of the two major dust sources, the Taklimakan Desert and the Gobi Desert, are marked in Figure

250    1.

Two simulations were conducted for the numerical test. One control run (CTRL) was conducted without dust and one
test run (DUST) was conducted with dust, both using the newly–implemented GOCART–Thompson microphysics
scheme. The GOCART aerosol model was applied to simulate aerosol processes (Ginoux et al., 2001;Ginoux et al.,
2004). Shao's dust emission (Kang et al., 2011;Shao et al., 2011) with soil data from the United states Geological
Survey (Soil Survey Staff, 1993), which have been demonstrated to have good performance in reproducing dust
emissions over East Asia were used to generate dust emission in the test run. The new wet scavenging scheme was
used for in-cloud wet scavenging of aerosols due to microphysical processes. Other important physical and chemical
parameterizations applied for the simulations are as follows. The Mellor–Yamada–Janjic (MYJ) turbulent kinetic
energy scheme was used for the planetary boundary layer parameterization (Janjić, 2002, 1994); the moisture
convective processes were parameterized by the Grell-Freitas scheme (Grell and Freitas, 2014); the short-wave (SW)
and long-wave (LW) radiation budgets were calculated by the Rapid Radiative Transfer Model for General Circulation
(RRTMG) SW and LW radiation schemes (Mlawer et al., 1997;Iacono et al., 2008), the gravitational settling and



surface deposition were combined for aerosol dry deposition calculation (Wesely, 1989); a simple washout method
was used for the below-cloud wet deposition of aerosols; and the aerosol optical properties were calculated based on
the volume-averaging method. As no dust emission is produced in CTRL, the setting for ice nucleation process in
CTRL followed the dust-free case in the original aerosol-aware Thompson scheme. The background concentration of
ice nuclei was set to be 1/L for the freezing of super-cooled water droplets into cloud ice, which was accounted by the
Bigg's parameterization (Bigg, 1953); the initiation of ice nucleation was determined by a temperature-dependent
function following Cooper (Cooper, 1986) when the temperature was below –20 °C, and $RH_i$ exceeded 105%
(Thompson and Eidhammer, 2014).

**4 Observations**
**4.1 Surface $PM_{10}$ observations**
The hourly observations of surface $PM_{10}$ concentration at ten environmental monitoring stations located in or
surrounding the dust source areas in East Asia were used to examine the capability of the model in reproducing the
trend and magnitude of dust levels at the ground surface during the simulation period. The ten stations were located
in Jinchang, Gansu Province, Yinchuan, Qinghai Province, Shizuishan, Ningxia Province, Baotou, Inner Mongolia,
and Yan'an, Shaanxi Province. The location of the ten stations are indicated by the blue dots in Figure 1.

**4.2 AERONET AOD observations**
The AERONET program is a ground-based aerosol remote sensing network for measuring aerosol optical properties
at sites distributed around the globe. This program provides a long-term database of aerosol optical properties such as
aerosol extinction coefficient, single-scattering albedo, and aerosol optical depth (AOD) measured at various
wavelength. The AOD represents the total amount of aerosols within the atmospheric column. The observational data
from two sites were available for comparison with the simulation results during the simulation period in this study.
One was Dalanzadgad located to the north of the Gobi Desert in Mongolia, and the other was the Semi-Arid Climate
and Environment Observatory of Lanzhou University (SACOL) located at Lanzhou, Gansu Province, China. The exact



locations of the two AERONET sites are depicted by the red triangles in Figure 1. All of the measured data had passed
the quality control standard level 2, with an uncertainty of ±0.01 (Holben et al., 2001).

**4.3 Satellite data**
**4.3.1 Multi-angle Imaging SpectroRadiometer (MISR)**
The MISR instrument aboard the Terra platform of the United State National Aeronautics and Space Administration
(NASA) has been monitoring aerosol properties globally since 2000. It observes the aerosol properties in four narrow
spectral centered at 443 nm, 555 nm, 670 nm, and 865 nm, due to which the aerosol properties even over highly bright
surfaces, such as deserts, can be retrieved (Martonchik et al., 2004;Diner et al., 1998). In this study, the AOD data at
555 nm retrieved from the MISR level 3 products with a spatial resolution of 0.5° were used for comparison with the
spatial distribution of simulated AOD over East Asia during the investigated period.

**4.3.1 Moderate Resolution Imaging Spectroradiometer (MODIS)**
The MODIS instruments aboard Terra and Aqua platforms of NASA monitor Earth's changes and provide global
high-resolution cloud and aerosol optical properties at a near-daily interval (Kaufman et al., 1997).
To retrieve aerosol information over bright surfaces, such as deserts, the Deep Blue algorithm was developed to
employ retrievals from the blue channels of the MODIS instruments, at which wavelength the surface reflectance is
very low, such that the presence of aerosol can be detected by increasing total reflectance and enhanced spectral
contrast (Hsu et al., 2006). By applying this algorithm, the AOD values at wavelengths of 214 nm, 470 nm, 550 nm,
and 670 nm over bright surfaces can be retrieved. In this study, the MODIS level 2 AOD data at a 550 nm with a
spatial resolution of 10 km were used for comparison with the simulated AOD during the simulation period.
Furthermore, MODIS combines infrared and visible techniques to detect physical and radiative cloud properties, and
a near-infrared algorithm was applied to retrieve the precipitable water vapor, including liquid and ice water content,
in the atmosphere (Gao and Kaufman, 1998). The thermal column water vapor path was then derived by integrating
the moisture profile throughout the atmospheric column. In this study, the ice water path retrieved from the MODIS





level 3 cloud products with a spatial resolution of one degree was used for comparison with simulated ice water path
during the simulation period.

**4.3.2 Cloud-Aerosol Lidar and Infrared Pathfinder Satellite Observation (CALIPSO)**
The Cloud-Aerosol Lidar and Infrared Pathfinder Satellite, which is aboard the Aqua platform of NASA, combines
an active Light Detection and Ranging (LIDAR) instrument with passive infrared and visible imagers to probe the
vertical structure and properties of thin clouds and aerosols around the globe (Vaughan et al., 2004). It aims to fill
existing gaps in the ability to measure the global distribution of aerosols and cloud properties, and provides three-
dimensional perspectives of how clouds and aerosols form, evolve, and affect weather and climate. It observes high-
resolution vertical profiles of aerosol and cloud extinction coefficient globally at wavelengths of 532nm and 1064 nm.
The atmospheric ice water content (IWC) is derived from the observational cloud extinction coefficients at 532 nm
(Winker et al., 2009). In this study, the vertical profiles of CALIPSO IWC with a horizontal resolution of 5 km and
vertical resolution of 60 m were applied to verify the performance of the model in simulating the vertical distribution
of atmospheric ice water content.

**5 Results and model validation**
**5.1 Dust over East Asia**
The time series of daily average dust load over the entire East Asia region (domain 1) during the simulation period is
shown in Figure 2a. In total four dust events occurred during the simulation period, lasting from March 18 to 25,
March 30 to April 7, April 9 to 19, and April 22 to 29, 2012. The case from April 22 to 29 was the most significant
one, with daily s dust mass load of double the other cases. The fraction of daily dust load for each size bin is also
shown in Figure 2a. The dust particles in the third, fourth and fifth bins with effective diameters ranging from 3.6 to
20 μm account for the major part of the total mass of dust aerosols, and dust particles with diameters smaller than 3.6
μm account for a minor fraction of the total mass of dust aerosols.





The time series of the daily average number density of dust particles over East Asia during the simulation period
shown in Figure 2b shows a similar distribution as that for dust load; the noteworthy distinction between the time
series for dust load and number density lies in the fraction of each size bin. The two size bins with the smallest
diameters (no larger than 3.6 μm) account for over 80% of the total number of dust particles, and the particles with
diameters smaller than 6 μm account for over 95% of the total number of dust particles, indicating that the smallest
dust particles are the main source of ice-friendly aerosol to serve as ice nuclei in the atmosphere.

**5.1.1 Surface PM$_{10}$ concentration**
To evaluate the performance of WRF-Chem in reproducing dust emission over East Asia, the simulated surface PM$_{10}$
concentration were compared with observations from ten environmental monitoring stations located near dust sources
and downwind areas. The time series of the observed and simulated surface PM$_{10}$ concentrations at the ten stations
during the simulation period are shown in Figure 3. Overall, the model shows a good performance in simulating the
dust cycle at different stations, with the trend and magnitude of the daily mean PM$_{10}$ concentration well captured at
most of the stations, although the surface PM$_{10}$ concentration was overestimated at one station in Jinchang during the
simulation period (Figure 3e), and the dust event on April 26 was also overestimated at the stations in Shizuishan
(Figure 3c and d) and Yinchuan (Figure 3i and j).
The performance statistics for the simulated results are shown in Table 1. The model tends to produce lower surface
PM$_{10}$ concentrations than those observed, as no other emissions were considered in the simulations. The mean bias
(MB) ranged from $-108.73$ μg/m$^3$ to 72.46 μg/m$^3$, with a mean over all of the stations of $-18.84$ μg/m$^3$. The mean
error (ME) ranged from 46.07 μg/m3 to 155.83 μg/m3, with a mean over all of the stations of 107.24 μg/m$^3$. The root
mean squared error (RMSE) ranged from 64.78 μg/m$^3$ to 317.73 μg/m$^3$, with a mean over all of the stations of 181.28
μg/m$^3$. The relatively large values of the MB, ME and RMSE are mainly attributed to the fact that no other aerosol
emissions were considered in the simulations other than dust, while the surface PM$_{10}$ concentration at the monitoring
stations is influenced by aerosols emitted from other sources, such as anthropogenic emissions. The correlation
coefficient (r) ranged from 0.59 to 0.87, with an average for all of the stations of 0.70. The comparisons between the
observed and simulated surface PM$_{10}$ concentration indicates that the model is capable of reproducing the surface dust
concentration reasonably during dust events over East Asia.




### 5.1.2 AOD time series

To examine the performance of the model in reproducing the column sum of dust in the atmosphere, the simulated
AOD values were compared with observations measured at two AERONET sites during the simulation period, as
shown in Figure 4.
The site at Dalanzadgad (Figure 4a) is located in Mongolia to the north of the Gobi Desert. Despite the fact that the
simulated AOD was overestimated at the end of March and in mid-April compared to the observed values, the trend
and magnitude of the AOD time series at Dalanzadgad was reasonably reproduced by the model during the simulation
period.
SACOL (Figure 4b) is a site located in Lanzhou, Gansu Province, which is a typical downwind area for dust in China.
Apart from the overestimation on April 23, the model showed a good performance in reproducing the time series of
AOD at SACOL during the entire simulation period, with the trend and magnitude of AOD well captured.

### 5.1.3 Satellite-observational AOD

The spatial distribution of mean simulated AOD during the simulation period was also compared with observed values
from MODIS and MISR products. The AOD observed by MODIS showed high values at the dust source region in
both March and April of 2012, as shown in Figures 5a and b. The region with high AOD values in the west part of the
circled area is the Taklimakan Desert, and the region with relatively lower AOD in the east part of the circled area is
the Gobi Desert. The mean observed AOD over the Gobi Desert was lower than that over the Taklimakan Desert in
both March and April, and the mean observed AOD was higher in April than in March over both dust source areas.
The spatial patterns of AOD observed by MISR are similar to MODIS, with comparable values over the Gobi Desert,
but significantly lower values over the Taklimakan Desert in both March and April (Figure 5c and d).
The spatial patterns for the mean simulated AOD were similar to the observed values for the observations in both
months, as shown in Figures 5e and f. The model shows a good capability in capturing the spatial characteristics of
the AOD, as well as the trend in AOD from March to April over the dust source areas. For example, the mean AOD



389 in the southern part of the Taklimakan Desert was higher than that in the northern part in March, and the mean AOD

390 showed an increase from March to April over the Gobi Desert, both of which were captured by the model. The values

391 of the mean simulated AOD over the Gobi Desert are comparable to the observational values from both MODIS and

392 MISR, but the mean simulated AOD over the Taklimakan Desert are between the values of the MISR observations

393 and the MODIS observations.

394 In summary, it was demonstrated that the dust emissions simulated by WRF-Chem are reliable for further analysis by

395 the comparison between the simulation results and the observations for surface $PM_{10}$ concentrations, as well as the

396 temporal and spatial distributions of AOD values.

397

398 **5.2 Atmospheric ice water content**

399 Dust particles are effective ice nuclei and play an important role in ice nucleation in the atmosphere under appropriate

400 conditions. With the large number of ice nuclei served by dust particles emitted into the atmosphere, an increase in

401 the number of ice crystals is expected in the results from DUST compared with those from CTRL, after taking into

402 account the effects of dust particles in the GOCART–Thompson microphysics scheme, as the ice nucleation process

403 is triggered by dust particles at appropriate temperature and relative humidity, as shown by the overall comparison

404 between the simulated cloud ice mixing ratio and ice crystal number concentration at each simulated data point (at

405 all model grids at hourly intervals) from CTRL and DUST during the entire simulation period in Figure 6.

406 As expected, the model produces a much higher cloud ice mixing ratio (Figure 6a) and ice crystal number

407 concentration (Figure 6b) in DUST. The simulated cloud ice mixing ratio produced in CTRL is lower than 2 μg/kg at

408 most data points during the simulation period, whereas the data points with simulated ice mixing ratio higher than 2

409 μg/kg are substantially increased in the output of DUST. Similarly, the simulated ice crystal number concentration

410 produced in CTRL is lower than $0.5 \times 10^6$ #/kg at most data points during the simulation period, by contrast, the

411 simulated ice crystal number concentration number concentration is higher than $0.5 \times 10^6$ #/kg at over a half of total

412 data points in DUST. The substantial increase of simulated cloud ice mixing ratio and ice crystal number concentration

413 indicates that the enhancement of ice nucleation process induced by dust is successfully reproduced by the newly-

414 implemented GOCART-Thompson microphysics scheme during the simulation period.



The spatial distributions of the simulated ice water path and ice crystal number density from CTRL and DUST in
Figure 7 further demonstrate the spatial pattern of the enhancement in cloud ice due to dust over East Asia. The ice
water path produced by CTRL was lower than 1 $g/m^2$ over the entire East Asia Region (Figure 7a). After considering
the effect of dust in the ice nucleation process, the ice water path produced by DUST increased substantially over the
entire region, especially over dust sources and downwind areas, with values as high as 10 $g/m^2$ (Figure 7b and c). The
ice water path was increased by one order of magnitude as a result of the effect of dust particles serving as ice nuclei
in the atmosphere. As shown in Figures 7d–f, the spatial pattern for the enhancement of ice crystal number density
over East Asia was similar with that for the ice water path. The ice crystal number density was increased by one order
of magnitude over vast areas of East Asia upon considering the effect of dust in the ice nucleation process in the
simulation.

**5.2.1 Ice water path**
The mean simulated ice water path during the simulation period was compared to the observed ice water path retrieved
from the MODIS products, as shown in Figure 8. MODIS observed ice water content in the atmosphere including
precipitating ice, such as snow and graupel, and cloud ice, which remains suspended in the upper troposphere and
lower stratosphere (Eliasson et al., 2011). Therefore, the simulated ice water path for comparison with the observations
also contained all of the precipitable ice variables output by the model, including snow, graupel, and cloud ice.
In Figure 8a, high ice water paths were observed at three areas over East Asia, as indicated by the red rectangles: in
the west part of the simulation domain containing the Taklimakan Desert and the Tibetan Plateau, South China, and
the area from Northeast China to Japan, with the highest values located at the south side of the Himalayas, and in
South China.
The simulated ice water paths produced by CTRL and DUST, as shown in Figures 7b and c, have nearly identical
spatial patterns and magnitudes. Typically, the ratio of cloud ice to the total column of ice particles in model outputs
is on the order of 0.1 to 0.3 (Waliser et al., 2009), and the effect of dust in ice nucleation leads to a direct enhancement
in cloud ice in the simulation, but not precipitable ice; therefore, the enhancement in cloud ice induced by dust is
concealed in the ice water path containing all atmospheric ice water, including precipitating ice and cloud ice.



Nevertheless, in the results of both CTRL and DUST, the model produced a high ice water path over West China, and
the concentrated ice water path from Northeast China to Japan was also captured, albeit with a different pattern.
However, the high ice water path over South China was missed in the results of both CTRL and DUST. This might
be attributable to the following reasons. Firstly, the temperature in the upper troposphere produced by the model might
be too high over this area for super-cooled water droplets freezing into ice particles. Secondly, the high ice water path
observed over South China is mainly due to strong convection motions that occur frequently over this area during
spring. The moisture convection produced by the model might be too weak to transport sufficient water vapor into the
upper troposphere, leading to an overestimation of the ice water path over this area.
The ice water path over areas other than the aforementioned regions in the simulation domain was underestimated by
the model; however, it can be seen that the ice water path produced in DUST was higher than that produced in CTRL
over the dust source region and Southwest Pacific, which is more consistent with the observed results, suggesting that
the model exhibits a better performance in reproducing ice water path over East Asia when the effect of dust is taken
into account in the simulation.

**5.2.2 Ice water content during dust events**
The vertical profile of the simulated ice water content was also compared with the observation from CALIPSO during
dust events. As mentioned in section 5.1, a total of four dust events occurred during the simulation period, lasting
from March 18 to 25, March 30 to April 7, April 9 to 19, and April 22 to 28, 2012. As shown in Figures 9 and 11, the
simulated vertical profiles of the ice water content during each dust events were compared with observations measured
at 06 UTC on March 21, 18 UTC on April 1, 18 UTC on April 9, and 05 UTC on April 23, 2012, when the orbit of
the satellite passed over East Asia.
Unlike the MODIS ice water path, CALIPSO measures the global distribution of aerosol and cloud properties by
LIDAR, which uses a laser to generate visible light with a wavelength of 1 μm or less to detect small particles or
droplets in the atmosphere. Therefore, CALIPSO instruments are more sensitive to tenuous ice clouds and liquid
clouds composed of small particles or droplets, which are invisible to instruments using signals of near-infrared or
infrared wavelength to detect clouds. Moreover, the LIDAR signal is attenuated rapidly in optically dense clouds that





the infrared or near-infrared signals can easily penetrate(Winker et al., 2010). As a result, the CALIPSO observations
of ice water content are  mostly points where the temperatures is lower than −40 °C and the altitude is greater than 6
km poleward to 12 km equatorward, and mostly those without precipitating ice. Given the above considerations, the
simulated ice water content profiles compared with the CALIPSO observations are referred to as only cloud ice in this
section.
The simulated dust load over East Asia at 06 UTC on March 21, 2012 is shown in Figure 9a, in which the dust covered
vast areas from West to East China between 35°N and 45°N, and the orbit of the satellite passed through the area with
heavy dust load at around 100°E. Along the satellite orbit, the abundant dust particles were transported to as high as
10 km aloft (Figure 9c). At this time, a high concentration of ice water content was observed along the satellite orbit
at an altitude of around 10 km between 30°N and 45°N (Figure 9e). The simulation result from CTRL (Figure 9g)
shows that the model produces some ice cloud at altitude of 9–10 km between 35°N and 45°N, but with much lower
ice water content compared to the observations. Nevertheless, by applying the GOCART–Thompson microphysics
scheme, the effect of dust in ice nucleation process was considered in DUST, leading to a much higher ice water
content at altitude of 9–10 km between 35°N and 45°N (Figure 9i), which is much more consistent with the
observations. The comparison between the simulation results from CTRL and DUST indicates that the high ice water
content observed by the satellite between 30°N and 35°N might be unrelated to microphysical processes, but instead
due to strong convective motions over South China.
On April 1, 2012, Central to East China was covered by a thick dust plume, and the orbit of the satellite passed through
areas with heavy dust load between 25°N and 43°N along 120°E at 18 UTC (Figure 9b). Dust particles were distributed
vertically from the surface to over 8 km along the satellite orbit (Figure 9d). A band of high ice water content was
observed by the satellite at altitude of 5 km to 10 km between 33°N and 44°N (Figure 9f), which was barely reproduced
in the results of the CTRL run without dust. In contrast, the observed band of high ice water content was reproduced
by the model in DUST with much more consistent location and magnitude (Figure 9j).
At 18 UTC on April 9, 2012, the satellite was scanning the east coast of China, which was covered by a thick dust
plume between 35°N and 45°N (Figure 10a), with dust particles lifted up to 10 km above the surface (Figure 10c).
High concentration of ice water content was observed by the satellite at altitude from 5 km to 11 km between 30°N
and 45°N (Figure 10e). In this case, the model reproduced the high concentration of ice water content at the observed





location in the results from both CTRL and DUST, although the ice water content was significantly underestimated in
the results from CTRL (Figure 10g), while it was well reproduced in the results from DUST (Figure 10j).
Similar to the previous cases, the satellite was scanning along east coast of China at 05 UTC on April 23, 2012, when
a dust plume was arriving in East China and affecting areas between 35°N and 45°N (Figure 10b), and dust particles
were distributed vertically from the surface to 10 km along the scanning track of the satellite (Figure 10d). Along the
orbit of the satellite, two bands with high ice water content were observed at altitudes between 5 km and 12 km, one
is located between 30°N and 37°N, and the other is located between 40°N and 45°N (Figure 10f). In the results from
CTRL, the model reproduced the bands of high ice water content at the correct locations, but with substantially lower
values (Figure 10h); however, upon taking into account the effect of dust in the GOCART-Thompson microphysics
scheme, the bands of high ice water content were well reproduced by the model, with much more consistent values
(Figure 10-j).
By comparing the satellite-observational and simulated vertical profiles of ice water content during the various dust
events, it was demonstrated that the newly-implemented GOCART–Thompson microphysics scheme reproduces the
enhancement of ice water content clouds in the mid- to upper troposphere by taking in to account the effect of dust in
the ice nucleation process, which substantially improves the simulation of cloud ice.

**5.2.3 Mean vertical profiles of ice water content**
The mean profiles of the observed ice water content, as well as the simulated ice water content from CTRL and DUST
for the four dust events discussed in Section 5.2.2, are shown in Figure 11. Note that the "mean profile" of ice water
content is the average over the available data points for the ice water content along the orbit of the satellite between
30°N to 45°N for each of the dust events shown in Figures 9 and 11.
The black lines in Figure 11 represent the mean profile of the observed ice water content, and the blue and red lines
represent the mean profiles of the simulated ice water content from CTRL and DUST, respectively.
Compared with the results from CTRL, the simulation for the vertical profile of the ice water content was substantially
improved in DUST for each dust event, with the enhancement of the ice nucleation process well captured by the



GOCART-Thompson microphysics scheme, although there were still discrepancies between observations and the
simulation results from DUST.
For the cases on March 21 and April 1, the peaks of ice water content were observed at 9.5 km and 8 km, respectively,
whereas the simulated peak of ice water content were located at 8 km and 7.5 km, respectively, with lower peak values.
The lower peak value for the case on March 21 was due to the missing of the high ice water content observed between
30°N to 45°N in the simulation results (Figure 9e and i), while the lower peak value for the case on April 1 was due
to the underestimation of the ice water content around 35°N (Figure 9f and j). The locations of the peaks of simulated
ice water content for the cases on April 9 and April 23 are more consistent with the observed peaks, but still possessed
lower values due to the missing or underestimation of high ice water content in the observations.

### 529     5.3 Sensitivity test and discussion

As discussed in Section 5.2.3, the simulation of cloud ice is greatly improved by considering the enhancement of ice
nucleation process induced by dust, which is well captured by the GOCART–Thompson microphysics scheme.
However, the ice water content is still underestimated by the model during dust events. To determine the reason for
this limitation, numerical experiments were performed to investigate the sensitivity of simulated ice water content to
the parameters for the ice nucleation parameterization in the GOCART–Thompson microphysics scheme.

### 536     5.3.1 Calibration factor $c_f$

The calibration factor $c_f$ is an empirical tuning coefficient derived from observational data from field and laboratory
experiments. It ranges from 1 to 6, and recommended to be 3 (DeMott et al., 2015), which was applied in the previous
simulations. An experiment was conducted to investigate the sensitivity of the simulated ice water content to $c_f$ values
ranging from 3 to 6.
The mean profiles of ice water content from simulation results were compared with the CALIPSO observations for
the dust events discussed in Section 5.2.2 and 5.2.3, as shown in Figure 12. For the cases on March 21 and April 1,
changing $c_f$ did not result in an increase of ice water content; instead, the simulated ice water content remained





consistent for $c_f$ values varying from 3 to 6. As ice nucleation occurs only in a super-saturated atmosphere with respect
to water vapor, an upper limit was set in the Thompson microphysics scheme, in that once the coagulation makes the
relative humidity in the atmosphere lower than the threshold relative humidity, which was set to 105% in the
simulations, the ice nucleation process is terminated. The consistency in simulated ice water content with increasing
$c_f$ indicates that the ice water content reaches the upper limit with all the available water vapor coagulated into ice
crystals when $c_f$ is equal to 3, lowering the relative humidity in the atmospheric column to below 105% for these two
cases.
For the case on April 9, the simulated ice water content increased between 6 km and 9 km and matched the observed
profile better when $c_f$ was equal to 4 and 5; however, when $c_f$ was set to 6, the simulated ice water content was lower
than that obtained with $c_f$ values of 4 or 5, although it matched the observed profile better than that produced with a
$c_f$ of 3.
For the case on April 23, two peaks were observed in the profile of simulated ice water content, located at 7 km and
10 km. The simulated ice water content remained unchanged with $c_f$ values varying from 3 to 6 for the peak at 10 km,
but increased upon changing the $c_f$ from 3 to 4, and remained the same upon changing the $c_f$ from 5 and 6 for the
peak at 7 km. In this case, the peak of the simulated ice water content at 7 km should correspond to the observed peak
between 6 km to 8 km, which was slightly overestimated by the model, and increasing the $c_f$ resulted in even larger
overestimation of this peak.
Given the above discussion, increasing the calibration factor $c_f$ from 3 to 6 does not necessarily lead to a significant
variation in the simulated ice water content during dust events, and the model achieves a relatively better performance
in reproducing the profile of ice water content when the $c_f$ is set to 3 or 4.

**5.3.2 Threshold RH$_i$**
In this study, the threshold RH$_i$ that triggers the ice nucleation process in the simulation was set to be 105%. Since the
ice water content is underestimated in the simulations, a sensitivity experiment was carried out to investigate the
response of simulated ice water content to a lower threshold RH$_i$ (100%).





The mean profiles of ice water content from the simulation results were compared with the CALIPSO observations
for the aforementioned dust events, as shown in Figure 13. With the threshold $RH_i$ lowered to 100%, the simulated
ice water content showed an increase throughout the vertical profile, with the most significant increase at the peak,
suggesting more consistency with the observations for all of the dust events, except the one on April 1. In the case on
April 1, the simulated ice water content increased at lower layers than the peak, but slightly decreased right at the peak
upon lowering the threshold relative humidity with respect to ice to 100% for the case. Overall, the simulation of ice
water content during dust events was significantly improved by lowering the threshold $RH_i$ from 105% to 100%.

**6 Conclusions**
The GOCART–Thompson scheme was implemented into WRF-Chem to couple the GOCART dust model and the
aerosol-aware Thompson microphysics scheme. By applying this microphysics scheme, the effect of dust on the ice
nucleation process by serving as ice nuclei in the atmosphere can be quantified and evaluated by the model
simultaneously with dust simulation. Numerical experiments, including a control run without dust and a test run with
dust, were then carried out to evaluate the performance of the GOCART–Thompson microphysics scheme in
simulating the effect of dust on the content of cloud ice over East Asia during a typical dust-intensive period, by
comparing the simulation results with various observations.
Based on the GOCART aerosol model the model reproduces dust emission reasonably well, by capturing the trend
and magnitude of surface $PM_{10}$ concentration at various environmental monitoring stations and the AOD at two
AERONET sites. The spatial patterns of the mean AOD over East Asia during the simulation period were also
consistent with satellite observations.
The effect of dust on the ice nucleation process by serving as ice nuclei was then quantified and evaluated in the
GOCART–Thompson microphysics scheme. Upon considering the effect of dust in the simulation, the simulated ice
water mixing ratio and ice crystal number concentration over East Asia were one order of magnitude higher than those
simulated without dust, with the most significant enhancements located over dust source regions and downwind areas.
By comparing the mean simulated ice water path over East Asia during the simulation period with MODIS
observations, it was demonstrated that the ice water path including cloud ice and precipitating ice is reasonably



reproduced by the model over most areas of East Asia, with the results from the simulation run with dust more
consistent with the observations.
Comparison between the vertical profiles of the satellite-observed and simulated ice water content during various dust
events and the entire simulation period further indicated that the enhancement of cloud ice induced by abundant dust
particles serving as ice nuclei is well captured by the GOCART–Thompson microphysics scheme, with the results
from the simulation with dust much more consistent with the satellite–observations.
Sensitivity experiments revealed that the simulated ice water content is not very sensitive to the calibration factor in
the DeMott2015 ice nucleation scheme, but the model delivered a slight better performance in reproducing the ice
water content when the calibration factor was set to 3 or 4. However, the simulated ice water content is sensitive to
the threshold $RH_i$ to trigger the ice nucleation process in the model, and the simulation of ice water content is
significantly improved upon lowering the threshold $RH_i$ from 105% to 100%.

**Acknowledgement**. We would like to acknowledge the provision of the MODIS and the MISR observations by the
Ministry of Environmental Protection Data Center, U.S. National Center for Atmospheric Research (NCAR), and the
CALIPSO data by the U.S. National Aeronautics and Space Administration (NASA) Data Center. We thank the
principal investigators and their staff for establishing and maintaining the two AERONET sites used in this study. The
AERONET data were obtained freely from the AERONET program website (https://aeronet.gsfc.nasa.gov/). We
appreciate the assistance of the Hong Kong Observatory (HKO), which provided the meteorological data. Lin Su
would like to thank Dr. Georg Grell, Dr. Stuart McKeen, and Dr. Ravan Ahmandov from the Earth System Research
Laboratory, U.S. National Oceanic and Atmospheric Administration for insightful discussions. Other data used this
paper are properly cited and referred to in the reference list. All data shown in the results are available upon request.
This    work    was    supported    by    NSFC/RGC    Grant    N_HKUST631/05,
NSFC-FD Grant U1033001, and the RGC Grant 16303416.



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



**List of tables and figures**


Table 1: Performance statistics for the model in simulating surface $PM_{10}$ concentrations at environmental monitoring
stations during the simulation period.
Figure 1: Nested domain set for the simulations. Blue dots represent the weather stations used for model validation.
TD: the Taklimakan Desert; GD: The Gobi Desert.
Figure 2: Time series of spatially averaged daily dust mass load (a) and daily number density of ice-friendly aerosol
(b) over East Asia (domain 1) during the simulation period.
Figure 3: Time series of hourly observed and simulated surface $PM_{10}$ concentrations at various environmental
monitoring stations.
Figure 4: Time series of daily mean observed and simulated aerosol optical depths at Dalanzadgad (a) and SACOL
(b).
Figure 5: Spatial distributions of monthly mean AOD from MODIS observations (a, b), MISR observations (c, d), and
simulation results (e, f) for March (left panel) and April (right panel) of 2012.
Figure 6: Simulated cloud ice mixing ratio (a) and cloud ice crystal number concentration (b) at each data point from
CTRL and DUST.
Figure 7: Spatial distributions for the temporal mean simulated cloud ice water path (a-c) and ice crystal number
density (d-f) from CTRL (left panel), DUST (middle panel), and the difference between CTRL and DUST (right panel)
over East Asia (domain 1) during the simulation period.
Figure 8: Spatial distribution for the mean ice water path (a) from MODIS observations, and the simulation results of
CTRL (b) and DUST (c) during the simulation period.
Figure 9: Spatial distribution for simulated dust load and satellite scanning track (a, b), the simulated vertical profile
of ice-friendly aerosol (GNIFA) number concentration (c, d), the CALIPSO vertical profile of ice water content (e, f),
and the simulated vertical profile of ice water content from CTRL (g, h) and DUST (i, j) for the case on March 21 (left
panel) and April 1 (right panel) of 2012.
Figure 10: As Figure 9 but for the cases on April 9 (left panel) and April 23, (right panel) of 2012.



Figure 11: Vertical profiles for the mean observed ice water content from CALIPSO, and the simulated ice water
content from CTRL and DUST for dust events on March 21, April 1, April 9, and April 23, 2012.
Figure 12: Vertical profiles for the mean observed ice water content from CALIPSO, and the simulated ice water
content with various $c_f$ for the dust events on March 21, April 1, April 9, and April 23, 2012.
Figure 13: Vertical profiles for the mean observational ice water content from CALIPSO, and the simulated ice water
content with threshold RH of 105% and 100% with respect to ice for the dust events on March 21, April 1, April 9,
and April 23, 2012.





**Table 1:** Performance statistics for the model in simulating surface $PM_{10}$ concentrations at environmental monitoring stations during the simulation period.

| STATION | LOCATION | MB | ME | RMSE | r |
|---------|----------|-----|-----|------|---|
| XCNAQ77 | BAOTOU | -36.18 | 80.43 | 94.88 | 0.59 |
| XCNAQ79 | | -10.05 | 75.83 | 106.58 | 0.62 |
| XCNAQ346 | SHIZUISHAN | 72.46 | 121.18 | 317.73 | 0.79 |
| XCNAQ347 | | 17.64 | 147.95 | 294.71 | 0.75 |
| XCNAQ340 | JINCHANG | -108.73 | 109.09 | 128.56 | 0.77 |
| XCNAQ342 | | -18.65 | 46.07 | 64.78 | 0.70 |
| XCNAQ335 | YAN'AN | -38.93 | 99.05 | 149.44 | 0.68 |
| XCNAQ336 | | -60.15 | 124.74 | 166.89 | 0.60 |
| XCNAQ344 | YINCHUAN | 33.97 | 112.26 | 240.27 | 0.87 |
| CN_1487 | | -39.62 | 155.83 | 249.00 | 0.62 |
| AVERAGE | | -18.84 | 107.24 | 181.28 | 0.70 |

MB: mean bias; ME: mean error; RMSE: root mean squared error; r: correlation coefficient.





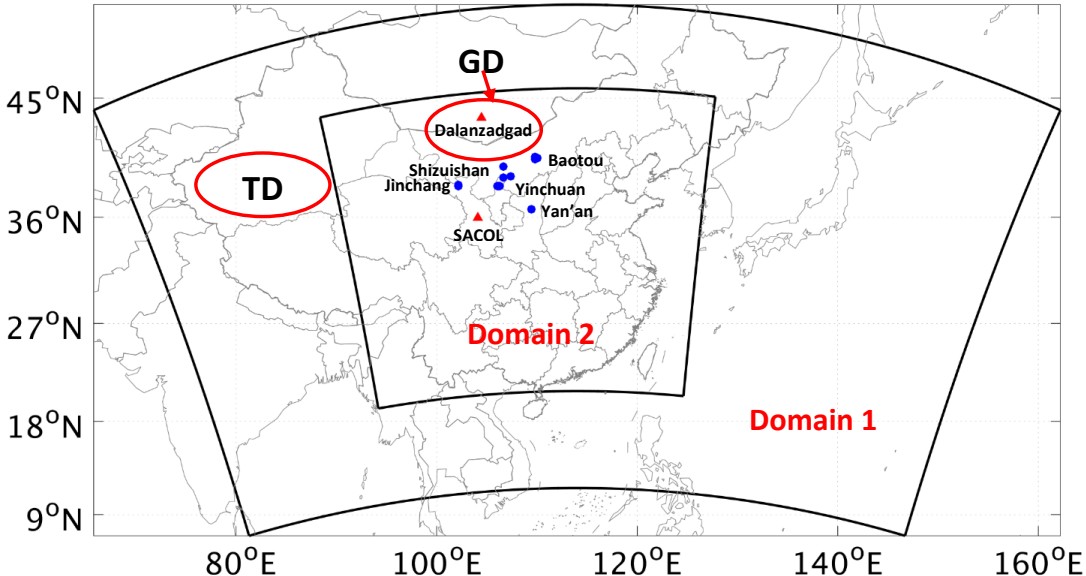

**Figure 1**: Nested domain set for the simulations. Blue dots represent the weather stations used for model validation. TD: the Taklimakan Desert; GD: The Gobi Desert.



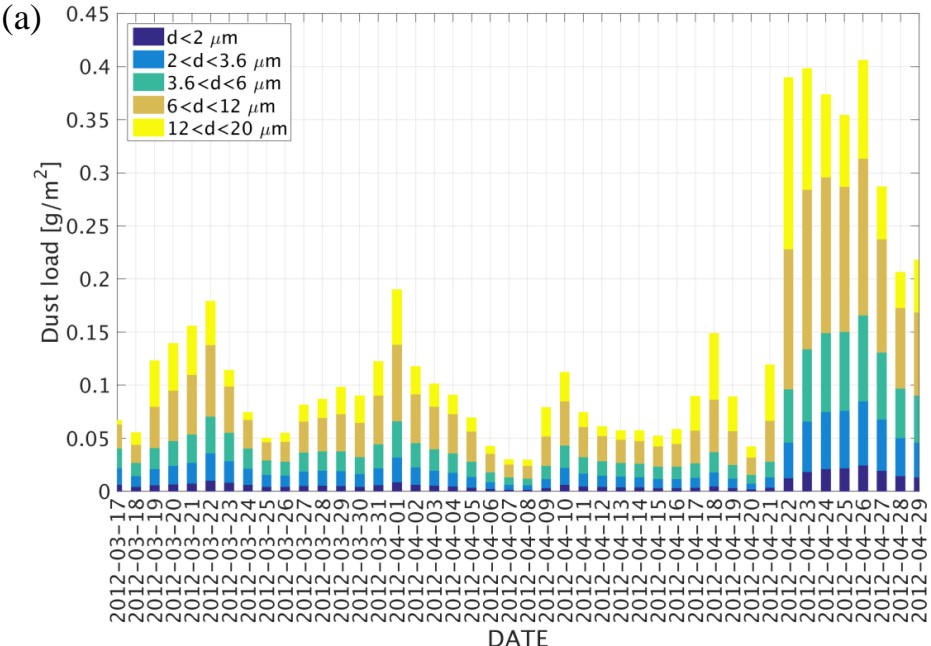

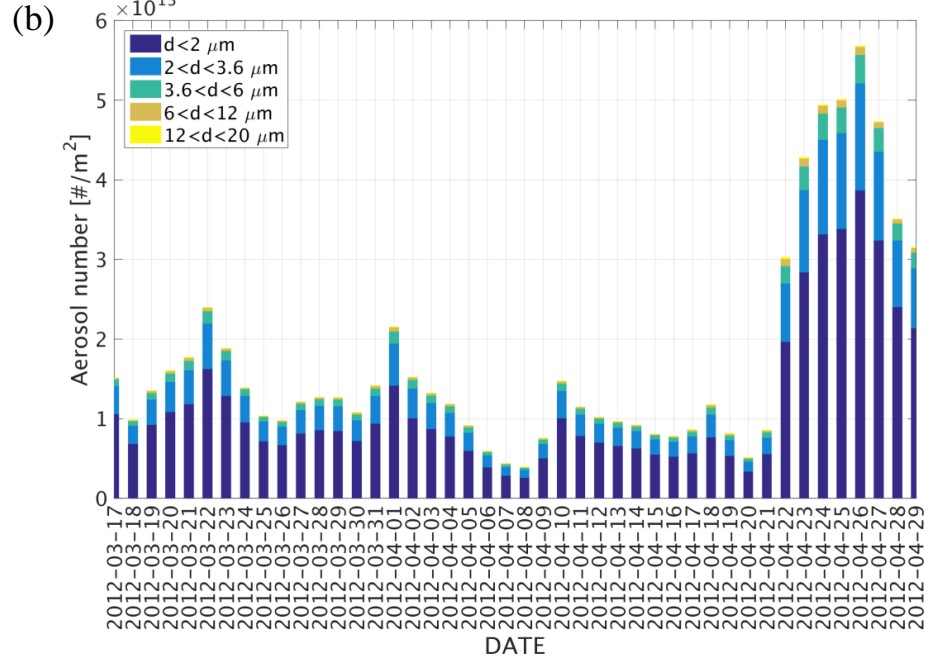

**Figure 2**: Time series of spatially averaged daily dust mass load (a) and daily number density of ice-friendly aerosol (b) over East Asia (domain 1) during the simulation period.





**Figure 3**: Time series of hourly observed and simulated surface $PM_{10}$ concentrations at various environmental monitoring stations.



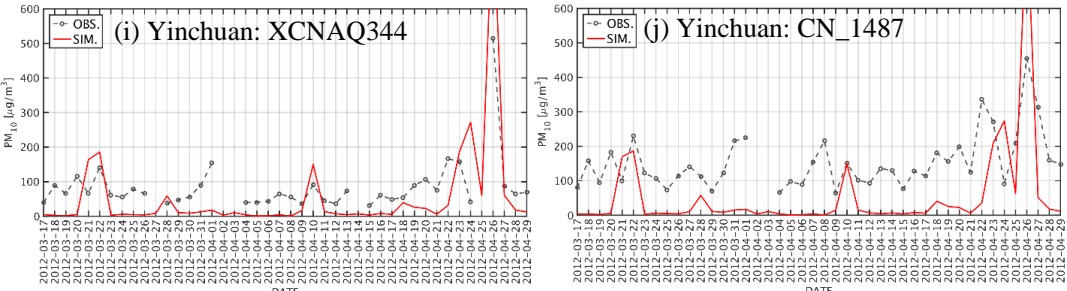

**Figure 3**: Continued.

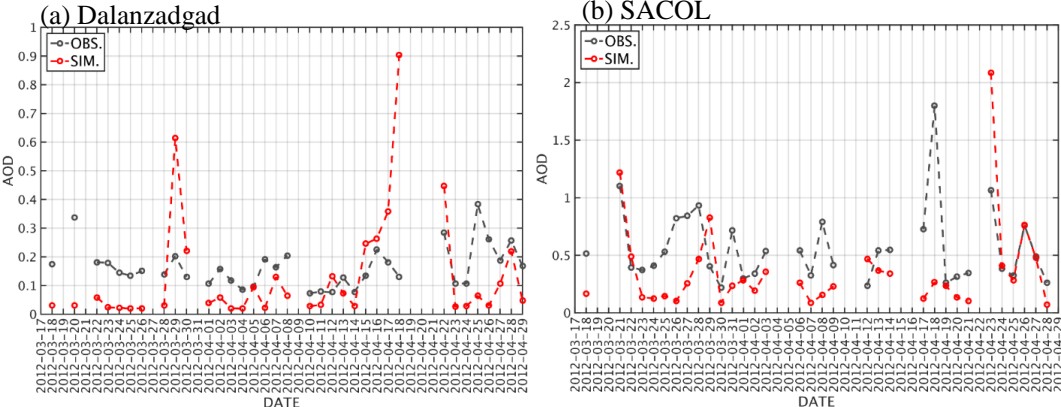

**Figure 4:** Time series of daily mean observed and simulated aerosol optical depths at Dalanzadgad (a) and SACOL (b).





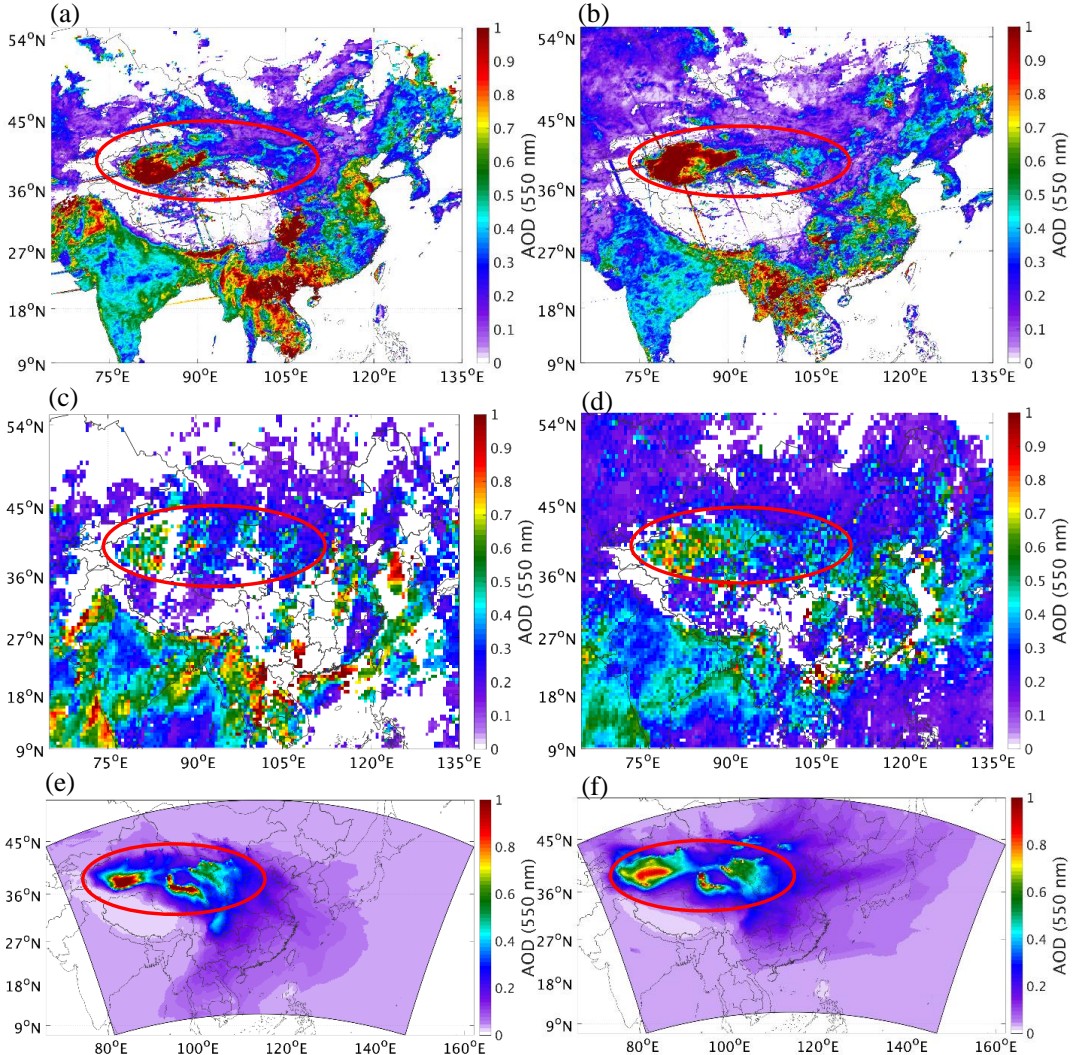

**Figure 5:** Spatial distributions of monthly mean AOD from MODIS observations (a, b), MISR observations (c, d), and simulation results (e, f) for March (left panel) and April (right panel) of 2012.





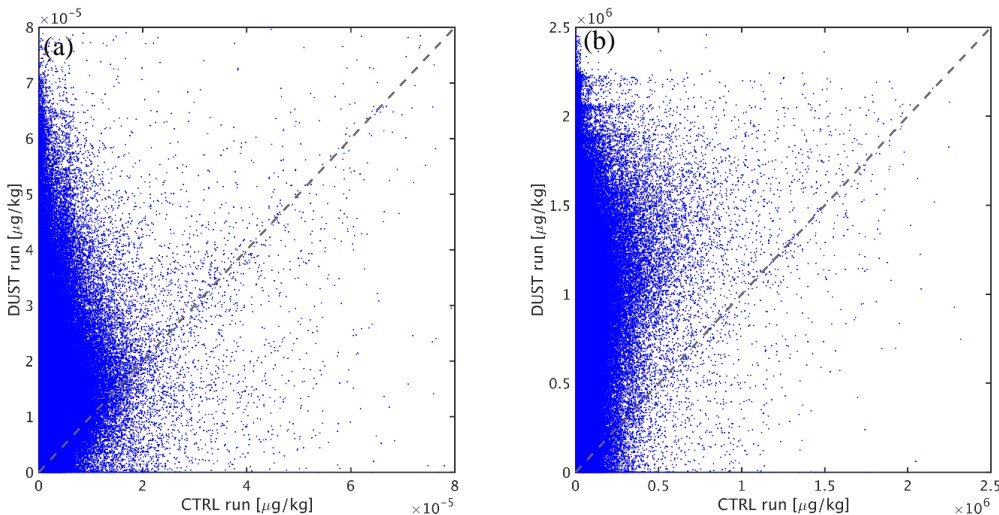

**Figure 6:** Simulated cloud ice mixing ratio (a) and cloud ice crystal number concentration (b) at each data point from CTRL and DUST.



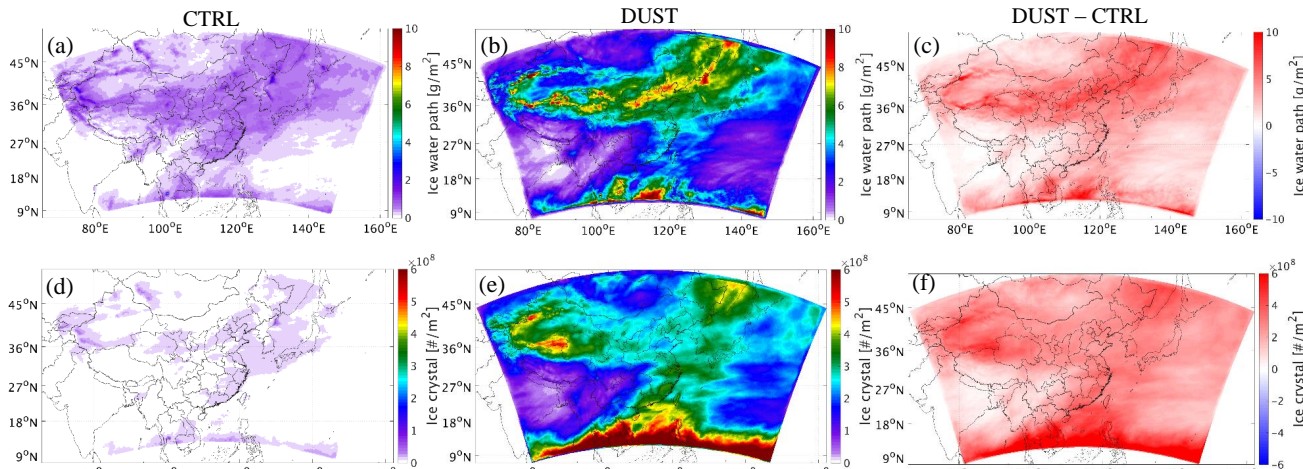

**Figure 7:** Spatial distributions for the temporal mean simulated cloud ice water path (a-c) and ice crystal number density (d-f) from CTRL (left panel), DUST (middle panel), and the difference between CTRL and DUST (right panel) over East Asia (domain 1) during the simulation period.



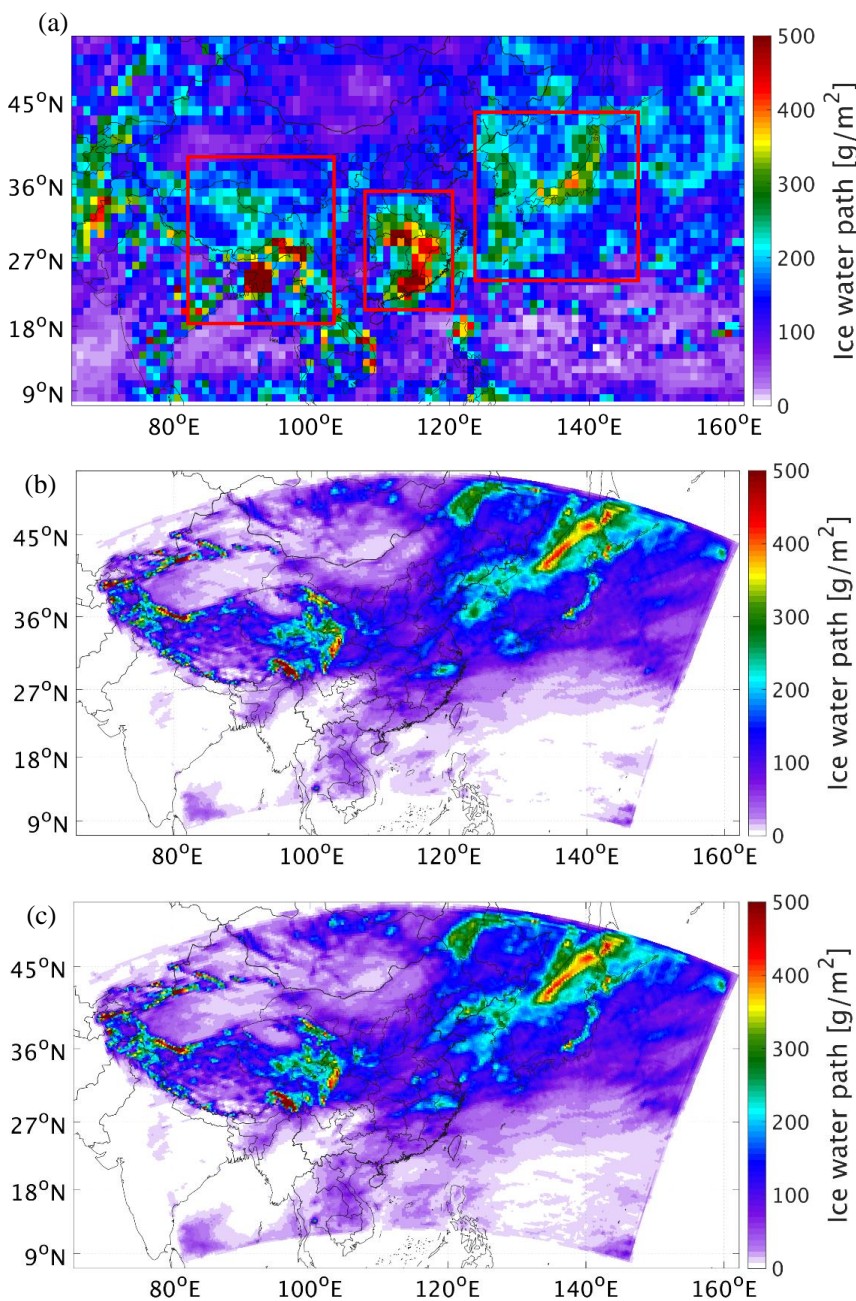

**Figure 8:** Spatial distribution for the mean ice water path (a) from MODIS observations, and the simulation results of CTRL (b) and DUST (c) during the simulation period.



**Figure 9:** Spatial distribution for simulated dust load and satellite scanning track (a, b), the simulated vertical profile of ice-friendly aerosol (GNIFA) number concentration (c, d), the CALIPSO vertical profile of ice water content (e, f), and the simulated vertical profile of ice water content from CTRL (g, h) and DUST (i, j) for the case on March 21 (left panel) and April 1 (right panel) of 2012.

**Figure 10:** As Figure 9 but for the cases on April 9 (left panel) and April 23, (right panel) of 2012.





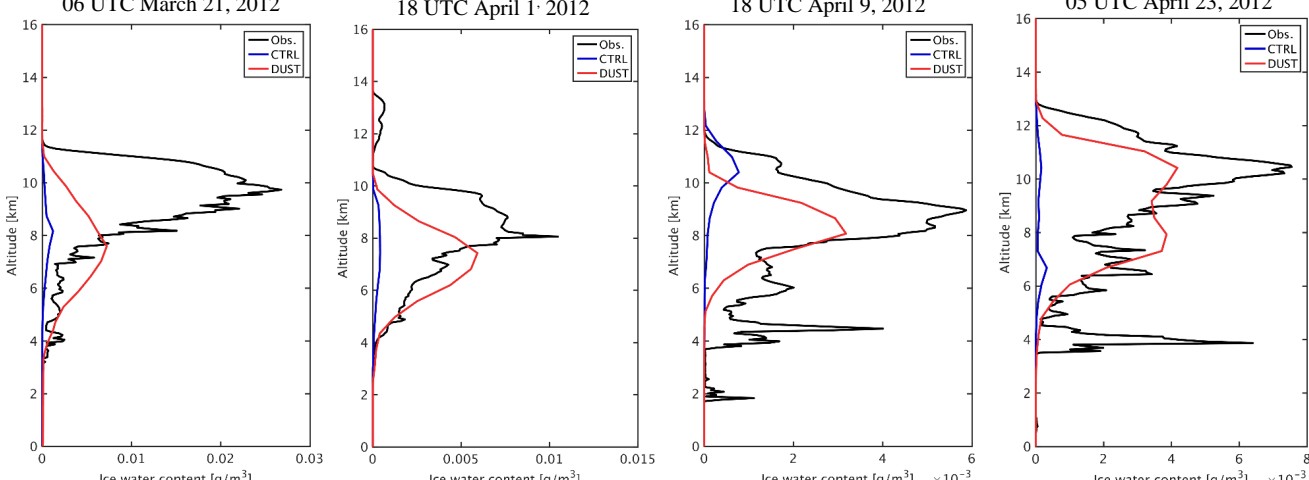

**Figure 11:** Vertical profiles for the mean observed ice water content from CALIPSO, and the simulated ice water content from CTRL and DUST for dust events on March 21, April 1, April 9, and April 23, 2012.





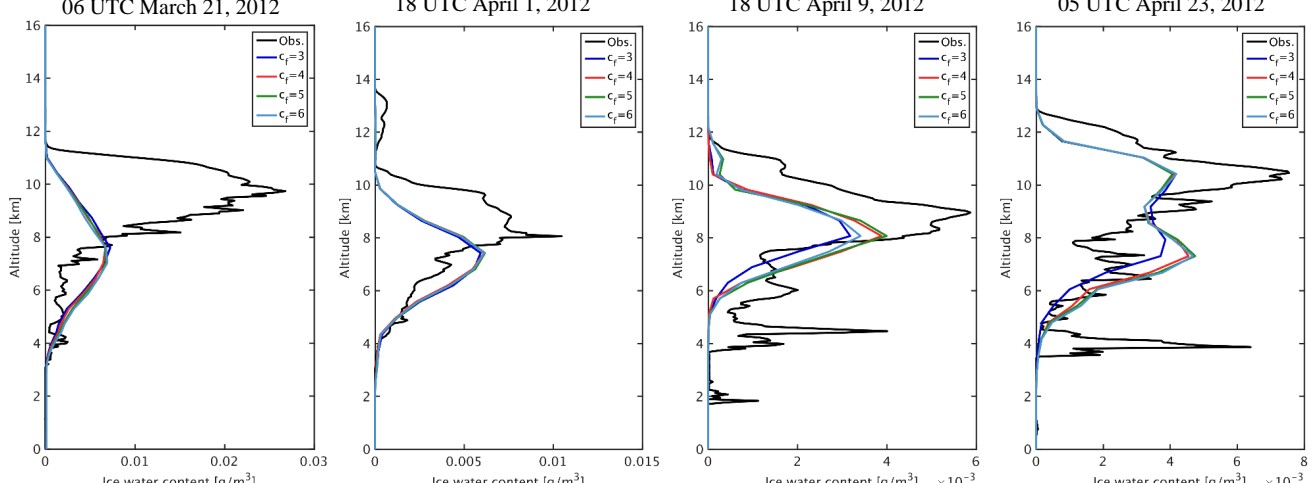

**Figure 12:** Vertical profiles for the mean observed ice water content from CALIPSO, and the simulated ice water content with various $c_f$ for the dust events on March 21, April 1, April 9, and April 23, 2012.



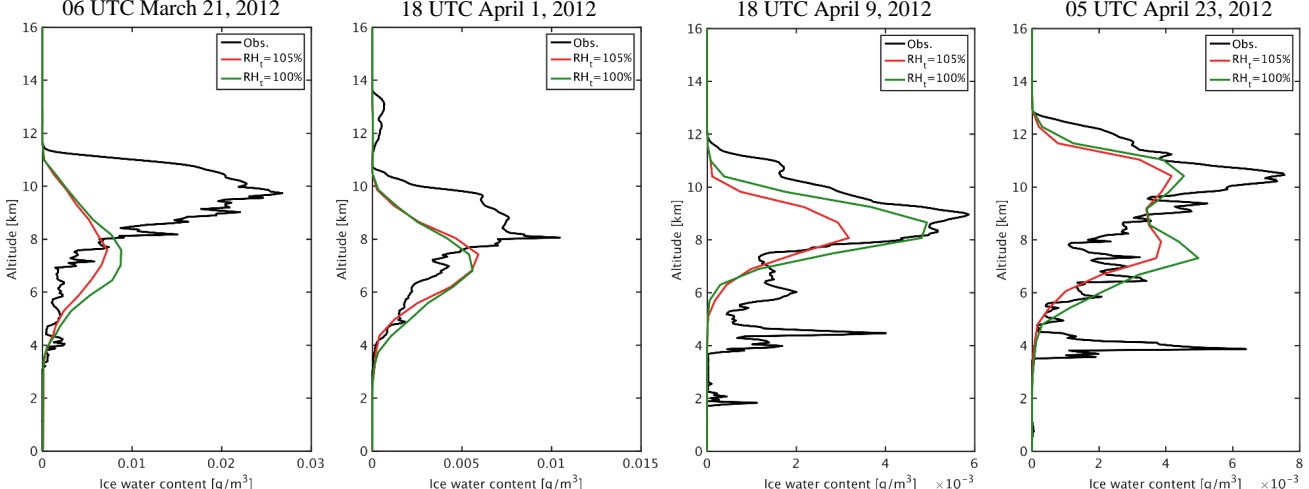

**Figure 13:** Vertical profiles for the mean observational ice water content from CALIPSO, and the simulated ice water content with threshold RH of 105% and 100% with respect to ice for the dust events on March 21, April 1, April 9, and April 23, 2012.