# Peer review of "Manuscript under review for journal Atmos. Chem. Phys."

_Atmospheric Chemistry and Physics, 2017_

## Referee Comment (RC1) · G. Thompson (Referee) · 7 Nov 2017

General comments:

I was aware of this work through various discussions with the first author while she was visiting Boulder, Colorado. Overall I find the manuscript reads quite well and has ample new material to warrant publishing. The results are clearly presented and the comparison to available observations is sufficient, although still a little less than I would prefer; however, this part of the world can be difficult to obtain high temporal and numerous spatial observations. Due to the nature of dust storms there are many times when a big event occurred but the observed data contain outages (noted in Figs. 3 and 4, for example). Ultimately I believe the paper requires relatively minor revisions at this stage.

Scientific issues:

In general, please state how much extra computational time is needed to run WRF-Chem as configured here as compared to a WRF simulation with all else the same including the aerosol-aware microphysics scheme but not utilizing WRF-Chem at all.

The reason I am asking is because we were also working on direct incorporation of an existing GOCART dust emissions scheme directly into WRF, but completely independent of using WRF-Chem. I conveyed this to the author in person a couple years ago. Since then, we also have the same contribution to the ice nuclei variable but simply combine 4 distinct dust sizes into the single category, final variable (e.g. QNIFA), external to WRF-Chem. The result is a far lower computation cost, probably by a massive amount.

As a test of our own newly added capability and due to seeing this manuscript, I ran the WRF model configured very similarly to what was done by these authors from 12 March to 30 April 2012. Solely as a quick-look graphical comparison to Fig. 4, I created a time series at the 2 AERONET sites for AOD obs versus the model. The resulting figure is attached to this review. The final AOD shown in the attached figure appears better at tracking the "background" value when dust storms are not present as compared with Fig. 4 in the manuscript, which has a model-predicted value that is nearly always lower than observations. When combining the data from both AERONET sites, the correlation coefficient of AOD for my WRF run was 0.59.

Besides stating directly in the paper what is the added cost of running WRF-Chem as compared to 'regular' WRF, would you please include the correlation coefficient for each data location shown in Figs. 3 and 4? A single value per panel would be nice to

see.

A point this test raises is whether or not the use of WRF-Chem is at all needed to gain a large computational efficiency while still predicting dust outbreaks and resulting changes to AOD, coverage of ice clouds, etc. Perhaps a few sentences in the revised manuscript to point out this alternate possibility could be made clear.

Minor comments:

At line#106 and similar lines mentioning the "Thompson scheme," would you please change all relevant places to "Thompson-Eidhammer scheme?" I would like to ensure proper credit to the coauthor of the WRF aerosol-aware scheme.

Line#118: you state: "In the current version of WRF-Chem..." then mention the number of water/ice-friendly aerosols. Actually, this is not technically correct as the creation of those 2 variables does not occur within any version of WRF-Chem. Those new variables are created irrespective of WRF-Chem for correctness in the paper.

Lines 126-128: "... can hardly represent realistic aerosol level..." is a bit too harsh. Actually, the use of a climatological aerosol concentration is closer to observations more often than not simply because the times when aerosols are extremely high or low as compared to climatology are the rarity in the true world as it takes a major weather change to produce the more extreme values. Take for example a place in the world with consistent southerly winds and a large urban region to the south of a point in question. When the winds are from the north, from a region of very low aerosols, then the point in question may be experiencing a large departure from climatology. Or if a significant weather front moves through a region and pushes away a majority of the aerosols, then the departure from average is more significant than the days in which the "regular weather regime" is occurring. So I am only suggesting to you that the wording here could be an exaggeration that using climo aerosols is always wrong.

Line 205 and 223: The internals of the Thompson-Eidhammer scheme contains a

wet deposition removal by precipitation. Did you disable this internal sink of aerosols when putting in your own wet removal process or is there a double-counting of aerosol removal?

Fig. 6b: I believe the units label is incorrect if the plot is supposed to show cloud ice number concentration.

Fig. 7: Can you offer any explanation for the very very large increase in ice number concentration showing along the southern boundary of the domain? The increase looks tremendous.

Line 433-434: If moist convection is too weak in the model and insufficient water vapor is not lofted high enough, then do you mean there is an *under* prediction of ice water path? The text says there is an over-prediction, but I do not understand this apparent contradiction.

Fig. 11: Is the peak shown $\sim$4km coinciding with the melting level? If so, is it a data processing issue of falsely identifying ice clouds in CALIPSO data?

Line 531: "...once the coagulation makes the relative humidity..." I cannot determine how the word coagulation is being used here. Can you please re-phrase this sentence?

Figs. 12 and 13: Is the ice water content computed from a combination of cloud ice mixing ratio and snow? I strongly advise combining both since cloud ice species in Thompson is literally only the smallest ice crystals (generally mean sizes below 40 microns) whereas the snow is much larger (mixing ratio as well as mean size).

Line 567: "A new microphysics scheme..." Calling this entirely a 'new' scheme is probably a stretch. The microphysics scheme is still much more than dust nucleating as ice, so a 'new treatment' for a source of dust used in the existing scheme is perhaps a more fair description.

[Figure]

AERONET & WRF AOD

---

## Author Comment (AC1) · 11 Dec 2017

We deeply thank Dr. Thompson for his valuable comments and suggestions, which help us improve the quality of the manuscript. It took us some time to submit the response, as we decided to revise the manuscript to address most of the comments before drafting the response. The detailed responses to each comment are shown as following.

Scientific issues: Computational time needed for the new treatment.

[Figure]

Response: As dust is produced by WRF-Chem for the new treatment of dust-ice interaction, it requires longer computational time than using standard WRF only. Generally, the speed ratio between standard WRF and WRF-Chem with the GOCART aerosol scheme is 5:3. In our opinion, it brings improvements either by directly incorporation a dust emission into WRF, or hooking up the GOCART aerosol scheme and microphysics scheme. It is more efficient in computing dust-ice interaction with the first option. However, there are several dust emission schemes in the GOCART aerosol scheme, and more will be added. These dust emission schemes might have different performances depending on the regions. Therefore, it allows users to apply different dust emission schemes to produce dust emission for evaluating dust-ice interaction by hooking up the GOCART aerosol scheme with microphysics scheme. Thanks very much for the information about your work, we will mention the extra computational time needed for the simulation, as well as the alternate possibility in the revised manuscript.

Statistics for the performance of the model in simulating AOD and PM10 concentration.

Response: The correlation coefficients for AOD is 0.46 and 0.65 for Dalanzadgad and SACOL, respectively. The statistics for the surface PM10 concentrations are displayed in Table 2, but we have also put the correlation coefficient of each station in Figure 3 and 4 in the revised manuscript.

Minor comments:

Comment 1: At line#106 and similar lines mentioning the "Thompson scheme," would you please change all relevant places to "Thompson-Eidhammer scheme?" I would like to ensure proper credit to the coauthor of the WRF aerosol-aware scheme.

Response: we have replaced "aerosol-aware Thompson scheme" with "Thompson-Eidhammer scheme" in the revised manuscript.

Comment 2: Line#118: you state: "In the current version of WRF-Chem..." then mention the number of water/ice-friendly aerosols. Actually, this is not technically correct

as the creation of those 2 variables does not occur within any version of WRF-Chem. Those new variables are created irrespective of WRF-Chem for correctness in the paper.

Response: We have revised the statement into "For the Thompson-Eidhammer scheme...".

Comment 3: Lines 126-128: "... can hardly represent realistic aerosol level..." is a bit too harsh. Actually, the use of a climatological aerosol concentration is closer to observations more often than not simply because the times when aerosols are extremely high or low as compared to climatology are the rarity in the true world as it takes a major weather change to produce the more extreme values. Take for example a place in the world with consistent southerly winds and a large urban region to the south of a point in question. When the winds are from the north, from a region of very low aerosols, then the point in question may be experiencing a large departure from climatology. Or if a significant weather front moves through a region and pushes away a majority of the aerosols, then the departure from average is more significant than the days in which the "regular weather regime" is occurring. So I am only suggesting to you that the wording here could be an exaggeration that using climo aerosols is always wrong.

Response: We have modified this part to make a fairer statement in the revised manuscript.

Comment 4: Line 205 and 223: The internals of the Thompson-Eidhammer scheme contains a wet deposition removal by precipitation. Did you disable this internal sink of aerosols when putting in your own wet removal process or is there a double-counting of aerosol removal?

Response: Yes. The GOCART aerosol scheme calculates the number concentration of aerosol every time step, it makes no sense to update it in the Thompson-Eidhammer scheme, so we turned it off while calculating the wet deposition using a new scheme.

Comment 5: Fig. 6b: I believe the units label is incorrect if the plot is supposed to show cloud ice number concentration.

Response: Revised in the updated manuscript.

Comment 6: Fig. 7: Can you offer any explanation for the very very large increase in ice number concentration showing along the southern boundary of the domain? The increase looks tremendous.

Response: During dust season, the outbreak of cold high system over northeast Asia can bring quantitative dust aerosol down to the South China Sea or even further, a typical case on March 26, 2012 is shown in Fig. R1 attached at the end of this response. In such cases, strong northwestlies swept across the entire China, and brought large amount of dust from source areas to the south border of the domain. Besides, the water vapor mixing ratio over south China Sea can be over five times as that over north China as shown in Fig. R2. Large amount of ice nuclei transported by winds, combining with abundant water vapor, results in a significant enhancement in the formation of ice crystals over the area.

Comment 7: Line 433-434: If moist convection is too weak in the model and insufficient water vapor is not lofted high enough, then do you mean there is an *under* prediction of ice water path? The text says there is an over-prediction, but I do not understand this apparent contradiction.

Response: It was a mistake and should be "underestimation", we have corrected it in the revised manuscript.

Comment 8: Fig. 11: Is the peak shown _4km coinciding with the melting level? If so, is it a data processing issue of falsely identifying ice clouds in CALIPSO data?

Response: The melting level is slightly below 4 km for the cases in Fig. 11, it is reasonable that ice clouds are observed at 4km, but the small peak at ∼2 km in the third case in Fig. 11 might be due to falsely identifying ice clouds in CALIPSO data.

[Figure]

Comment 9: Line 531: "...once the coagulation makes the relative humidity..." I cannot determine how the word coagulation is being used here. Can you please re-phrase this sentence?

Response: It has been revised in to " freezing of droplets" in the updated manuscript.

Comment 10: Figs. 12 and 13: Is the ice water content computed from a combination of cloud ice mixing ratio and snow? I strongly advise combining both since cloud ice species in Thompson is literally only the smallest ice crystals (generally mean sizes below 40 microns) whereas the snow is much larger (mixing ratio as well as mean size).

Response: The ice water content shown in Fig. 12 and 13 contains only cloud ice. The reason we did not use a combination of cloud ice mixing ratio and snow is that, the ice water content in CALIPSO products represents only the amount of cloud ice, as the CALIPSO instruments are more sensitive to ice clouds and liquid clouds composed of small particles or droplets, and the LIDAR signal emitted from CALIPSO attenuated rapidly in optically dense clouds (more detailed description can be seen in section 5.2.2). Therefore, for comparison of the vertical profile of ice water content with the CALIPSO products, we only apply the mixing ratio of cloud ice. The combination of simulated cloud ice mixing ratio and snow is compared to MODIS products in Fig. 8, but only for a spatial comparison, due to the limitation of observations.

Comment 11: Line 567: "A new microphysics scheme..." Calling this entirely a 'new' scheme is probably a stretch. The microphysics scheme is still much more than dust nucleating as ice, so a 'new treatment' for a source of dust used in the existing scheme is perhaps a more fair description.

Response: we have revised it into "a new treatment" to avoid overstating the implementation work.

[Figure]

2017.

[Figure]

**Fig. 1.** The spatial distribution of dust particle number density over East Asia at 04 UTC, March 26, 2012.

[Figure]

**Fig. 2.** The spatial distribution of the mean column sum of water vapor over East Asia during the simulation period.

---

## Referee Comment (RC2) · Anonymous Referee #3 · 12 Jan 2018

The authors present the coupling between the GOCART aerosol model and the aerosol-aware Thompson cloud microphysics scheme implemented in the WRF-Chem model. The newly-implemented GOCART-Thompson microphysics scheme is evaluated against observations (in terms of AOD, $PM_{10}$ concentrations, ice water path, and ice water content) and is used to simulate the indirect dust effect (showing the importance of dust in the enhancement of cloud ice) over East Asia during a dust-intensive period.

The paper is worth to be published, but only after a careful work of revision by the authors. The paper provides novel material. The comparison with the observations is well developed (the authors used several observational data, from ground stations and satellite). The structure of the paper and the presentation of the results are well thought, but the paper is spoiled by a poor exposition: sometimes the explanations are not complete, some expressions are not precise, there are some repetitions and some mistakes (e.g.: between singular/plural subject/verb, missing words, typos, etc.). Although it is understandable what the authors mean, I strongly recommend the authors to review again the manuscript and consider what written in the comments below.

**General comments**

It is a bit confusing the fact that the authors sometimes write that the improvements are due to the application of the newly-implemented GOCART-Thompson microphysics scheme (e.g. line 29) and other times that the improvements are due to the consideration of dust effect (e.g. line 508). In this work, all the simulations use the newly-implemented GOCART-Thompson microphysics scheme (as written at line 252) and the dust emissions are switched on in the DUST simulation while they are switched off in the CTRL one. As a consequence, if Fig. 8, 9, 10, 11 show that DUST is better than CTRL in comparison with the observations, the reason is the inclusion of dust effect in ice nucleation (not the application of the newly-implemented GOCART-Thompson microphysics scheme, although the dust effect in ice nucleation can be considered only thanks to the GOCART-Thompson microphysics scheme). Therefore, I would correct lines 29, 506-508 (while lines 452-453 are correct). Strictly speaking, the improvements due to the application of the newly-implemented GOCART-Thompson microphysics scheme could be demonstrated only by a comparison between the standard version of WRF-Chem and the "new" version WRF-Chem + the newly-implemented GOCART-Thompson microphysics scheme.

In CTRL results, are the ice crystals only produced by homogeneous nucleation? And what about IWP and IWC in Fig. 8, 9, 10, 11? Some words should be spent also to explain CTRL results, as they also show weak signals in correspondence to the dust events (although without dust emissions).

The advantages of using the newly-implemented GOCART-Thompson microphysics scheme should be stressed more in comparison to the standard version of WRF-Chem (e.g. in the section about the implementation). For instance, besides the possibility to consider the dust effect on ice nucleation, the coupling allows to consider it during particular episodes like intensive dust events. What about simulations with other aerosol species emissions?

As dust particles come from different sources (also anthropogenic), please, specify that here dust is actually "mineral dust" (at least at lines 22 in the abstract, 37, 42, ...), although this should be clear because "natural sources" are mentioned at the beginning of the Introduction.
Please, quantify (percentage?) how much dust contributes to the global aerosol burden at line 37.

Just as general comment, my thought is that CTRL simulation with no aerosol emissions at all is a rather "extreme" scenario. I would have preferred to keep more realistic aerosol emissions (including for example black carbon, organics, ...) and to reduce dust emissions (e.g. by 50%) in CTRL, instead of excluding them.

The authors should define more explicitly the direct, semi-direct and indirect effects of dust in the paragraph between the lines 42-48, because these effects are then commented in the next paragraph (from line 49). It would be nice also to add some lines (between 46-48) about the role of dust in ice nucleation, as it is the crucial process considered in the paper (heterogeneous nucleation, nucleation modes, thermodynamic conditions,...). Parts of lines 65-66 could be moved here.

It would be appropriate to develop the paragraph 93-96 by adding some information about the WRF model and the comparison between WRF and WRF-Chem (what can you do with the second one that you cannot do with only WRF? Or, why is it better to use WRF-Chem instead of WRF? In terms of aerosols and ice nucleation processes...). This addition is needed because I guess that the authors will speak about the WRF model after the requests by the first Referee.

Subsection 2.3 starts saying that the modifications made by the authors are three, but then 4 sub-subsections follow. Since the sub-subsections 2.3.2 describes how the ice nucleation is treated by the aerosol-aware Thompson scheme (and, if I am correct, there are no indications about modifications by the authors in 2.3.2) I would suggest to reorder the structure of the current section 2. I think it would be clearer for the reader first to get to know the ice nucleation scheme (what is written in 2.2 and 2.3.3), and then to focus on the implementation work done by the authors. Thus, it would be better to split section 2 in two parts:
Section 2: Model description, with 2.1 and 2.2 (= 2.2 + 2.3.3)
Section 3: Implementation of GOCART-Thompson microphysics scheme, with 3.1 (=2.3.1), 3.2 (=2.3.3), and 3.3 (=2.3.4).
Moreover, it would be appropriate to say explicitly what the authors mean with "GOCART-Thompson microphysics scheme" (at the beginning of the new section 3?).

It is not clear to me if the DeMott2015 scheme is already available in the aerosol-aware Thompson microphysics scheme or not. At line 121 the authors say that condensation and immersion freezing is parameterized by DeMott2010, while at line 192 they say that they apply DeMott2015. Please, clarify this.
Moreover, why do the authors well explain the schemes by DeMott and they do not describe the scheme by Phillips et al. 2008? Maybe they could add just few lines also for the latter scheme.

Just to be sure, the simulated data plotted in Fig. 3, 4, 5 belong to the test run DUST (not to CTRL), is it correct? Please, specify it.
In Fig. 5, the authors focus their attention on the circled area with dust sources. However, there is an evident difference between the simulated and the observed AOD values over India and South China. Could they explain why?

**Minor comments**

Minor comments are divided in: "Technical comments/Suggestions" (i.e. what has to be explained better, indications about the units, what should be added, etc.) and "Text corrections/suggestions" (i.e. some typos, mistakes or other suggestions regarding the form).

**Technical comments/Suggestions**

**Lines 30-33:** I would stop the first sentence after "scheme" at line 31, and I would join "Results suggest that..." with the last sentence.
Here, I would only mention that the ice nucleation scheme is not much sensitive to the calibration factor without specifying the numbers (I think this information is too specific for the Abstract).

**Lines 61-62:** Please, introduce here the abbreviations CCN and IN and use them in the rest of the text.

**Line 69:** Add reference after "WRF-Chem" or rephrase the sentence, otherwise it seems that GOCART has been implemented by the authors.

**Lines 74-77:** I would change the sentence to: "IN 2014, the aerosol-aware Thompson microphysics scheme, which takes into account the aerosols serving as CLOUD CONDENSATION NUCLEI AND ice nuclei, has been implemented into WRF, enabling the model to explicitly predict the number concentration for cloud droplets AND ICE CRYSTALS (Thompson and Eidhammer, 2014)." Would it be correct?

**Lines 82-85:** This paragraph should be developed and made clearer.
The first point is the implementation, the second point is the validation plus investigation.

Thus, it would be better to split the sentence in two parts and to mention also the validation of the model.

Please, explain what is the meaning of "online simulations" or write differently (probably it is better to rephrase the sentence as the expression "online simulations" does not appear again in the text, so it is not important to introduce it here).

Moreover, I would specify "THE GOCART aerosol model" at line 82: "we AIM to fully couple the aerosol-aware Thompson microphysics scheme with THE GOCART aerosol model in ...". I think it would be clearer since only the GOCART aerosol mode has been mentioned until this point and only later, at line 95, the reader will discover that GOCART is one of three schemes.

**Line 101:** Is there a reference about the implementation of GOCART into WRF-Chem to add here?

**Lines 104-105:** It is not clear the correspondence between the emission schemes and the list of references. Please, write: "Shao's dust emission scheme (REF) is one of the three dust emission schemes in the GOCART aerosol model. THE OTHER TWO SCHEMES ARE DEFINED BY REF-1 AND REF-2".

Otherwise, simply write: "Shao's dust emission scheme (REF) is one of the dust emission schemes in the GOCART aerosol model" without saying "three".

**Lines 113-116:** As far as I understood, the aerosol-aware version of the Thompson scheme is the evolution of the Thompson scheme. Please, add the reference for the Thompson scheme at line 113 and for the aerosol-aware Thompson scheme at line 116.

It would be better to write "... and therefore it explicitly predicts the number concentrations of cloud condensation nuclei and ice nuclei as well as the number concentration of cloud droplets and ice crystals ". Or, did I misunderstand the meaning?

**Lines 149-152:** Nicer to read: "... can be approximated through the mean effective radius ($r_{dust}$, UNIT) and density ($\rho_{dust}$, UNIT) OF DUST PARTICLES for that size bin:" deleting line 152.

Please, specify always the UNIT, e.g.: (A, UNIT). Also at line 159.

**Line 153:** More correct should be "DUST number concentration (N, $\#/kg$) for ..." instead of "The aerosol number concentration N ...". Also at line 156.

**Line 169:** According to the reference DeMott et al. 2010, $n_{ice,T_k}$ in equation (4) is actually $n_{IN,T_k}$ defined as number concentration of IN (instead of ice crystal number concentration). I would follow the notation and the variable descriptions of the reference.

The same is valid for equation (5).

Then, the authors could add that $n_{IN} = n_{ice}$ as generally IN are not enough to deplete supersaturation (if this were the case, the more efficient IN would nucleate first and $n_{ice}$ would be less than $n_{IN}$).

**Lines 173, 177:** Again, according to DeMott et al., I think it should be "... or low concentration of IN compared ..." and "... or higher concentrations of IN based ...", instead of "ice crystals".

**Lines 174, 181:** If the authors refer to Fig. S1 of DeMott2010, the relationship is between IN number concentrations and aerosol particle number concentrations (not ice crystals).

Similarly at line 181.

**Lines 201-209:** These two paragraphs describe how to implement the GOCART-Thompson scheme, but they are badly written (the explanations are not clear and some of them are redundant). The authors should check this sub-subsection and rewrite it more clearly.

**Lines 229-230:** Delete "for the following simulation". I mean: "The mass mixing ratio for dust aerosol in a particular size bin $n$ is then updated FOR the next time step $(t + 1)$:"

It would be more correct to use labels for the time in equation (8), e.g.: $C^{t+1} = C^t - wetscav^t$.

**Line 233:** The wet removal of dust is proportional to the concentration of what? Dust number concentration? Please, specify it.

Is there a reference to add for the wet deposition scheme in the GOCART aerosol model?

**Line 245:** Information about the model time step could be added.

**Line 256:** It is better to write here that no other aerosol emissions are considered besides dust (what written at line 358), so the reader knows this when the analysis starts.

Does it mean that ice nucleation in CTRL occurs only via homogeneous nucleation? Please, write it.

**Lines 264-265:** Are there some references for the washout method and the volume-averaging method?

**Lines 274-278:** This paragraph should be rearranged and $PM_{10}$ should be defined. Moreover, the word "trend" is not used with its statistical meaning (the authors do not compute any trend of dust concentration, rather they check the magnitude and the behaviour of the temporal series). For this reason it would be better to avoid the use of "trend" (also in the next text) and to use, for instance, behaviour, evolution, etc.

Possibility for this paragraph: "The observations of surface concentration of particulate matter with diameter $< 10 \ \mu m$ ($PM_{10}$) measured at ten environmental monitoring stations were used to examine the capability of the model in reproducing dust levels at the ground surface during the simulation period. The ten stations (indicated by blue dots in Figure 1) were located in or surrounding the dust source areas in East Asia: Jinchang, Gansu Province, Yinchuan, Qinghai Province, Shizuishan, Ningxia Province, Baotou, Inner Mongolia, and Yan'an, Shaanxi Province."

At the end the authors could add some characteristics of the measurements: hourly and which unit?

**Lines 334-336:** It seems to me that the third bin is more often comparable to the second one than to the fourth one, so I would mention here only "fourth and fifth".

Please, quantify "major part" and "minor fraction", as done for Fig. 2b (lines 340-341).

**Lines 337-342:** Is the number density vertically integrated? Please, specify it.

In Fig. 2, "ice-friendly aerosol" is written in the caption and "aerosol number" in the y-axis, while the text refers only to dust particles. The fact that dust particles are the only aerosols (ice-friendly aerosols) emitted comes out only later (line 358). It would be better to mention this already in the section of Model Configurations (and write "dust aerosol" in Fig. 2).

**Line 348:** Please, write that the time series of the simulated concentrations are extracted from the nearest grid point to the geographical coordinates of the stations. Is it correct?

**Lines 349-352:** This paragraph should be modified. I do not see that "the surface $PM_{10}$ concentration was overestimated at one station in Jinchang", the general tendency of the model should be considered before the individual events (the sentence in lines 353-354 could be moved here), and Fig, 3g and h should be also mentioned.

Thus, the paragraph could become: "Overall, the model shows a good performance in simulating the dust cycle at THE different LOCATIONS, with EVOLUTION and magnitude of the daily mean $PM_{10}$ concentration well captured at most of the stations. THE MODEL TENDS TO PRODUCE SURFACE $PM_{10}$ CONCENTRATIONS LOWER THAN THOSE OBSERVED, AS NO OTHER EMISSIONS WERE CONSIDERED IN THE SIMULATIONS. HOWEVER, the dust eventS on MARCH 21 AND April 26 wERE overestimated BY THE MODEL at the LOCATIONS in ... " c,d,g,h,i,j.

**Line 360:** To compute the correlation, which data are used? Hourly measurements and which simulated concentrations? Please, specify it.

**Lines 369-372:** This paragraph should be a bit modified. It would be better to say firstly the general performance of the model (remembering that the underestimation is due to the fact that there are no other emissions apart from dust) and to point out later the overestimations in the two periods.

Remember to avoid the usage of "trend".

The temporal means of simulated and observed AOD could be added.

The same considerations are valid for the next paragraph regarding SACOL.

**Lines 377:** In this sub-subsection both modeled results and observations are analysed, therefore, the title could be changed from "Satellite-observational AOD" to "AOD spatial distribution".

**Line 385:** Is it possible to quantify (percentage?) "lower values"?

**Line 388:** Actually, with a first look at the plots in Fig. 5 it does not seem that the model well reproduces the evolution from March to April because the AOD looks higher in March than in April (while the observations show the opposite, as written at line 383). I see that the motivations for such sentence are provided in the next lines but, please, make line 388 clearer (and more modest).

"trend" $\rightarrow$ "evolution".

**Lines 391-393:** Add some numbers (AOD means over GD and TD?).

**Lines 406-412:** Add the mean values of cloud ice mixing ratio and ice crystal number concentration averaged over the domain 1 (?) and the simulated period, for DUST and CTRL.

**Lines 421:** Write the explanation for the strong positive bias over the southern part of the domain in Fig. 7.

**Lines 437-440:** It is still not clear to me why CTRL and DUST show almost the same ice water path.

**Lines 451:** Difficult to appreciate that the ice water path over dust source regions is higher in DUST than in CTRL. Is it possible to add a temporal-spatial mean computed for this region?

**Line 459:** Please, specify if the simulated profiles refer to exactly the same time (e.g. 06 UTC, ...) of the observations or if they have been averaged (daily means?).

**Lines 468-469:** The sentence is not clear in my opinion. What do the authors mean with "points"?

**Lines 478-479:** As both CTRL and DUST use the newly-implemented GOCART-Thompson scheme (as written at line 252), I think this sentence is not technically correct: the higher IWC values are due to the fact that in DUST the effect of dust is considered (and not to the use of the GOCART-Thompson microphysics scheme). Is it correct? Like it is written at lines 530-531.

**Lines 505-508:** According to the same considerations written above, please, rephrase also this sentence.

**Line 520:** Before analysing the single cases, please, describe the main discrepancies: peaks of DUST are always lower in altitude and in magnitude.

**Lines 545-549:** Not well formulated.
Moreover, why is the calibration factor linked to the relative humidity? It is not clear to me the reasoning which leads the authors to their conclusion.

**Lines 551-554:** No profile matches the observations, better to write for instance: "... and WAS CLOSER TO the observed profile when ...".
"... set to 3 AND 6, ..".
Delete the part starting with "although": the difference between 3 and 6 is too small that I would not assert this.

**Line 563:** I would have said 4 or 5... Also at line 603 in the Conclusions.

**Line 600:** Please, add that the model generally underpredicts the IWC.

**Text corrections/suggestions**

**Line 20:** Better "... and THE aerosol-aware ..." ?

**Line 24:** I would switch the order because I think that the typical season is spring, not spring 2012: "... during spring, a typical dust-intensive season, in 2012."

**Line 26:** "increases by" → "increase UP TO".

**Line 28:** "demonstrated" → "demonstrateS" (the present tense is used in the Abstract).

**Lines 38-39:** Repetition of the word "major", possibly find a synonym.

**Line 42:** "Dust in the atmospherE ALTERS ... ".

**Line 43:** No comma after "atmosphere".

**Line 45:** No comma after "cloud".

**Line 46:** No comma after "nuclei".

**Line 49:** "To date, many studies ..." (with "m" in lower case).

**Line 54:** Better "... on dust-cloud interactionS ...".

**Line 60:** Better "... considering aerosol-cloud interactionS ...";
"... in regional MODELS, ...";
No comma after "schemes".

**Line 62:** No comma after "treated".

**Lines 59-63:** The sentence is too long in my opinion and not very clear in the second part.

**Line 75:** Remove "droplet" (it is specified in the next line).

**Line 80:** "... climate modelS, which ...".

**Line 90:** "... IN section 6."

**Line 94:** "... and the interactionS in between ...".

**Line 99:** "... sulfate, MINERAL dust, ...".

**Line 107:** No comma after "2016".

**Line 118:** "... number concentrationS using ...".

**Lines 118-123:** The sentence is too long in my opinion, the authors could separate the liquid part from the ice part.
Write in parenthesis only the year.

**Line 123:** "... water dropletS is ...".

**Line 128:** Repetition of the word "multiple", possibly find a synonym.

**Line 133:** Better "... aerosol-cloud interactionS ...".

**Lines 138-140:** Maybe not very clear: "modifications" (line 138) of what?
One possibility could be: "To investigate the real-time indirect effects of dust aerosol over East Asia, the GOCART model HAS BEEN COUPLED TO the aerosol-aware Thompson microphysics scheme. TO DO THIS, WRF-Chem version 3.8.1 HAS BEEN MODIFIED IN THREE STEPS: modification of ..." or something similar.

**Line 145:** "... the number concentrationS of aerosols are ...";
"... to evaluatE ...".

**Line 156:** Put "n" in italic style, like at line 153.

**Line 158:** No comma after "study".

**Lines 164-165:** I would move the reference to line 164: "(DeMott et al., 2010, hereafter DeMott2010 scheme)".

**Lines 166:** Delete "to account for condensation and immersion freezing" it is obvious from two lines before.

**Line 170:** Put "a, b, c, d" in italic style.

**Line 176:** Similarly to before, I would write: "(DeMott et al., 2015, hereafter the DeMott2015 scheme)".

**Lines 177-178:** Repetition of the word "latest", possibly find a synonym.

**Line 181:** "DeMott2015" (with "M" in upper case).

**Lines 185-187:** The last sentence could be deleted, there is nothing new with respect to the sentence at lines 176-178.

**Line 188:** "The number concentration of ice crystals produced by ..." without the word "that" and the singular form for "concentration" (otherwise, later, it should be: "is" → "are" and "that" → "those").

**Line 193:** Add comma after the first "scheme".

**Lines 210-212:** Move "is calculated" before: " ... at grid point (i,j,k) is calculated, I.E. the tendency of ...".

**Line 222:** "... the fraction of dust particle for each size bin ($\varphi$, UNIT) can be ...".

**Line 226:** "... the loss of dust mass due to the microphysical processes (*wetscav*, UNIT) for a particular size bin $n$ is ...".

**Lines 236-237:** In my opinion, it would be nicer to specify and describe the two experiments from the beginning, as the characteristics written in the following lines actually regard both of them and not only "A numerical experiment" as written at the start of the sentence. Therefore, I would move the lines 251-252 near to 236-237, e.g.: "TWO numerical experimentS WERE conducted to examine the performance of the newly-implemented GOCART-Thompson microphysics scheme in simulating the ice nucleation process induced by dust in the atmosphere. One control run (CTRL) was conducted without dust and one test run (DUST) was conducted with dust, both using the GOCART-Thompson microphysics scheme.". In this case, the first sentence at line 251 should be removed.

**Line 238:** Where? "... dust events in 2012 OVER EAST ASIA were ...".

**Line 239:** Remove "for this numerical test".

**Line 241:** The analysis has not started yet, therefore: "further" → "the".

**Line 246:** "simulationS".

**Lines 247-248:** No comma after "km".

**Line 249:** Specify here (TD) and (GD), as used in Fig. 1.

**Line 256:** Add comma after "East Asia".
Is "Shao's dust emission" the subject? If yes, "were" → "was".

**Line 262:** " , the gravitational..." → " ; the gravitational...".

**Lines 265-266:** The sentence about CTRL could be moved at line 256, so it is in contrast to DUST. I.e.: "... used to generate dust emission in the test run DUST. As no dust emission is produced in CTRL,...".

**Line 269:** Write in parenthesis only the year.

**Line 284:** Remove the sentence with the meaning of AOD. It is not necessary. Otherwise, it would be better to add the meanings also for the other quantities (aerosol extinction and single-scattering albedo).

**Lines 294, 321:** "observes" → "measures".

**Line 295:** "... spectral BAND centred at ...".

**Line 301:** Earth's changes of what? E.g.:?

**Line 303:** ", such as deserts," can be removed, because it is said before.

**Line 307:** "... at 550 nm..." (remove "a").

**Line 333:** Please, rephrase the sentence after the comma.

**Line 339:** Or simply: "... between the TWO time series lies in ...".

**Lines 345-347:** "To evaluate the performance of WRF-Chem in reproducing dust emissionS over East Asia, the simulated surface $PM_{10}$ concentrationS were compared with THE observations from THE ten environmental monitoring stations located near dust sources and downwind areas (DESCRIBED IN SUBSECTION 4.1)."
Delete "at the ten stations" at line 347.

**Lines 355-357:** Delete "of" after "all".

**Line 378:** "The spatial distribution of MONTHLY mean simulated AOD was compared with ...".

**Lines 379-382:** It would be nicer to explain firstly what the circled area indicates and to describe later the AOD values inside the circle, so exchange the order.

**Line 386:** Remove "for the observations".
Add that the similarity is stronger with MODIS.

**Line 389:** "... the mean OBSERVED AOD was higher in the southern part of the Taklimakan Desert than that in the northern part in March and showed an increase ...".

**Line 398:** Given the content of this subsection and the other sub-subsections, I would personally change the title to something like "Cloud ice over East Asia" (similarly to 5.1 Dust over East Asia).

**Lines 400-405:** Sentence too long and not well written. The part "as the ice nucleation process is triggered by dust particles at appropriate temperature and relative humidity," can be deleted, it is a repetition. A new sentence could then start as: "Figure 6 shows the overall comparison ...".

**Line 405:** No new line.

**Line 416:** Remove "spatial pattern of the".

**Line 427:** "The mean simulated ice water ..." → "The simulated ice water path AVERAGED over the simulation period ...". The same at line 593.

**Line 432:** Remove last "in".

**Line 434:** No comma after "Hymalayas".

**Line 439:** Remove "in the simulation,".

**Line 448:** "overestimation" → "underestimation".

**Lines 456-527:** Use sometimes "IWC" (defined at line 323) instead of "ice water content".

**Line 458:** Make reference to Fig. 9 and Fig. 10 (not to Fig. 11 which is described in the next sub-subsection).

**Line 485:** Remove "through areas with heavy dust load", it is not really so.

**Line 490:** It is not really "east coast"...

**Line 495:** "well" → "better".

**Line 497:** "in East China" → "from the dust sources".

**Line 497:** "in to" → "into".

**Line 514:** Fig. 9 and Fig. 10 (not Fig. 11).

**Lines 515-515:** Remove these lines, which belong to the caption of the figure.

**Line 517:** Remove "the simulation for".

**Line 526:** "are" → "were" (the authors have always used the past tense).

**Line 527:** "in" → "with respect to".

**Line 527:** "for" → "of".

**Lines 537-540:** Repetition of the word "from", possibly find a synonym somewhere.

**Line 539:** "THREE OTHER experimentS WERE conducted to investigate the ..." should be clearer.

**Lines 545-548:** Change the word "coagulation".

**Line 555:** "profile" → "profileS".

**Line 558:** Remove "at 7 km", it is not needed with "In this case".

**Line 559:** Remove the part after "model", it is a repetition of what written before.

**Line 571:** No comma after "profile".
"peak" → "peakS".

**Line 573:** "... at lower ALTITUDES than the OBSERVED peak...".

**Lines 573-574:** Please, rephrase the sentence after "but".

**Line 578:** "... to couple the GOCART AEROSOL model TO the ...".

**Line 579:** "By applying this NEWLY-IMPLEMENTED microphysics scheme, ...".

**Lines 580-581:** Remove "by the model simultaneously with dust simulation" or explain better.

**Lines 585-588:** "trend" → "evolution".
"... at THE LOCATIONS OF various monitoring stations ...".

**Line 589:** Remove "by serving as ice nuclei".

**Line 595:** "... reproduced by the model over most areas of East Asia, ALTHOUGH SLIGHTLY UN-
DERESTIMATED." and then start a new sentence.
Remove "run".

**Line 598:** Remove "and the entire simulation period further", it is not correct because the sentence
before refers to the IWC profiles during the dust events, along the satellite orbit or averaged
along it (Fig. 9, 10, 11).

**Line 601:** "... calibration factor DEFINED in the DeMott2015 ...".

**Lines 604:** Make it simpler: "... in the model AND IS significantly ...".

**Figures**

In Fig. 2, Fig. 3, Fig. 4, please, do not write the year 2012 in the tick-labels of the x-axis (to
make the plot "lighter"), rather write explicitly the simulation period in the captions.

**Fig.1:** - try to reduce the size of blue dots or draw the contours in order to show that there are 10
dots;
- in the caption write "Blue dots represent the TEN MONITORING stations used ...";
- in the caption add the meaning of the red triangles.

**Fig.3:** Why are there written only 5 different locations (2 per line), while in subsection 4.1 ten
locations are listed? It would be clearer to write the ten different locations (one per plot).
Otherwise, if there is a reason to group the stations, it would be better to specify it in
subsection 4.1.

**Table 1:** Same considerations as before (write 10 locations?).
Add the units of the quantities computed in the table.

**Fig.5:** - is it possible to use the same projections for all plots?
- make country lines a bit thicker;
- near the word AOD (along the color bars), "MODIS", "MISR" and "modeled" could be
added (as subscript?);
- the wavelength of MISR should be 555 instead of 550.

**Fig.6:** Wrong unit in 6b.

**Fig.8:** - is it possible to use the same projections for all plots?
- make country lines a bit thicker.

**Fig.9 and 10:** - increase the blank space between the two columns of plots;
- write the quantity (IWC) besides the unit (near the color bar);
- what do the red rectangles indicate? They are never used in the text for the explanations,
I think they could be removed;
- make country lines a bit thicker;
- It would be nice to plot also the orographic profile (to explain the white areas below the
GNIFA values).

**Fig.11,12,13:** - write "DURING the dust events ..";
- "MEAN vertical profiles OF THE observed ice water content from CALIPSO and the
simulated ice water content from ...".

---

## Author Comment (AC2) · 8 Feb 2018

We deeply thank the anonymous referee for the valuable and very detailed comments and suggestions. All the comments have been addressed and the responses to each comment are listed below.

General Comments:

Comment #1: It is a bit confusing the fact that the authors sometimes write that the improvements are due to the application of the newly-implemented GOCART-Thompson microphysics scheme (e.g. line 29) and other times that the improvements are due to the consideration of dust effect (e.g. line 508). In this work, all the simulations use the newly-implemented GOCART-Thompson microphysics scheme (as written at line 252) and the dust emissions are switched on in the DUST simulation while they are switched off in the CTRL one. As a consequence, if Fig. 8, 9, 10, 11 show that DUST is better than CTRL in comparison with the observations, the reason is the inclusion of dust effect in ice nucleation (not the application of the newly-implemented GOCART-Thompson microphysics scheme, although the dust effect in ice nucleation can be considered only thanks to the GOCART-Thompson microphysics scheme). Therefore, I would correct lines 29, 506-508 (while lines 452-453 are correct). Strictly speaking, the improvements due to the application of the newly-implemented GOCART-Thompson microphysics scheme could be demonstrated only by a comparison between the standard version of WRF-Chem and the "new" version WRF-Chem + the newly-implemented GOCART-Thompson microphysics scheme.
Response: We have modified the manuscript to clarify that the improvement in the simulation of ice water content was due to the inclusion of dust effect.

Comment #2: In CTRL results, are the ice crystals only produced by homogeneous nucleation? And what about IWP and IWC in Fig. 8, 9, 10, 11? Some words should be spent also to explain CTRL results, as they also show weak signals in correspondence to the dust events (although without dust emissions).
Response: Ice crystals can be produced by both homogeneous nucleation and heterogeneous nucleation in CTRL run. We have clarified it in section 3. In both CTRL and DUST, the newly–implemented GOCART–Thompson microphysics scheme was used for condensation and immersion freezing; the deposition nucleation is determined by the parameterization of Phillips et al. (Phillips et al., 2008), and the freezing of deliquesced aerosols using the hygroscopic aerosol concentration is parameterized following Koop et al. (Koop et al., 2000), with the background aerosol concentration set to be 1/L, which means that in CTRL, the background aerosol concentration is 1/L.

Comment #3: The advantages of using the newly-implemented GOCART-Thompson microphysics scheme should be stressed more in comparison to the standard version of WRF-Chem (e.g. in the section about the implementation). For instance, besides the possibility to consider the dust effect on ice nu-cleation, the coupling allows to consider it during particular episodes like intensive dust events. What about simulations with other aerosol species emissions?
Response: We have added a paragraph at the end of section 2 to stress the advantage of coupling the GOCART aerosol model with the Thompson-Eidhammer microphysics scheme.

Comment #4: As dust particles come from different sources (also anthropogenic), please, specify that here dust is actually "mineral dust" (at least at lines 22 in the abstract, 37, 42, ...), although this should be clear because "natural

sources" are mentioned at the beginning of the Introduction. Please, quantify (percentage?) how much dust contributes to the global aerosol burden at line 37.

Response: We have specified in the manuscript that only "mineral dust" was produced in the DUST simulation. The contribution of dust to the global aerosol burden has been added in Section 1.

Comment #5: Just as general comment, my thought is that CTRL simulation with no aerosol emissions at all is a rather "extreme" scenario. I would have preferred to keep more realistic aerosol emissions (including for example black carbon, organics, ...) and to reduce dust emissions (e.g. by 50%) in CTRL, instead of excluding them.

Response: Other than to evaluate performance of the newly-implemented treatment in simulating ice nucleation involving dust aerosol, we wanted to investigate the effects of dust on the regional weather system over East Asia (which was presented in part II of this paper), therefore, the CTRL run must be an "extreme" scenario without dust aerosol.

Comment #6: The authors should define more explicitly the direct, semi-direct and indirect effects of dust in the paragraph between the lines 42-48, because these effects are then commented in the next paragraph (from line 49). It would be nice also to add some lines (between 46-48) about the role of dust in ice nucleation, as it is the crucial process considered in the paper (heterogeneous nucleation, nucleation modes, thermodynamic conditions,...). Parts of lines 65-66 could be moved here.

Response: The second paragraph in Section 1 has been revised to explicitly the direct, semi-direct and indirect effects of dust.

Comment #7: It would be appropriate to develop the paragraph 93-96 by adding some information about the WRF model and the comparison between WRF and WRF-Chem (what can you do with the second one that you cannot do with only WRF? Or, why is it better to use WRF-Chem instead of WRF? In terms of aerosols and ice nucleation processes...). This addition is needed because I guess that the authors will speak about the WRF model after the requests by the first Referee.

Response: We have included the information about the comparison between WRF and WRF-Chem in Section 1: "Therefore, the aerosol-aware Thompson-Eidhammer scheme is an ideal microphysics scheme for evaluating the effect of dust in atmospheric ice nucleation processes. However, this scheme is not coupled with any aerosol model in WRF-Chem, the Weather Research and Forecast model coupled with Chemistry. When the aerosol-aware Thompson-Eidhammer microphysics scheme is activated, the model reads in pre-given climatological aerosol data derived from the output of other global climate models, which introduces large errors into the estimation of the effects of dust in microphysical processes. This problem can be solved by embedding a dust scheme into Thompson-Eidhammer scheme, or couple the microphysics scheme with WRF-Chem. Compared with WRF, WRF-Chem integrates various emission schemes and aerosol mechanisms for simulating the emission, transport, mixing, and chemical transformation of aerosols simultaneously with the meteorology (Grell et al., 2013). Therefore, WRF-Chem is more capable of producing a realistic aerosol field by comparing the performances of different emission schemes or aerosol mechanisms."

Comment #8: Subsection 2.3 starts saying that the modifications made by the authors are three, but then 4

sub-subsections follow. Since the sub-subsections 2.3.2 describes how the ice nucleation is treated by the aerosol-aware Thompson scheme (and, if I am correct, there are no indications about modifications by the authors in 2.3.2) I would suggest to reorder the structure of the current section 2.

I think it would be clearer for the reader first to get to know the ice nucleation scheme (what is written in 2.2 and 2.3.3), and then to focus on the implementation work done by the authors. Thus, it would be better to split section 2 in two parts: Section 2: Model description, with 2.1 and 2.2 (= 2.2 + 2.3.3) Section 3: Implementation of GOCART-Thompson microphysics scheme, with 3.1 (=2.3.1), 3.2 (=2.3.3), and 3.3 (=2.3.4). Moreover, it would be appropriate to say explicitly what the authors mean with "GOCART- Thompson microphysics scheme" (at the beginning of the new section 3?).

Response: We have re-ordered the subsections as suggested. The meaning of GOCART-Thompson microphysics scheme has been added at the beginning of section 3.

Comment #9: It is not clear to me if the DeMott2015 scheme is already available in the aerosol-aware Thompson microphysics scheme or not. At line 121 the authors say that condensation and immersion freezing is parameterized by DeMott2010, while at line 192 they say that they apply DeMott2015. Please, clarify this.

Moreover, why do the authors well explain the schemes by DeMott and they do not describe the scheme by Phillips et al. 2008? Maybe they could add just few lines also for the latter scheme.

Response: The default scheme for condensation and immersion freezing in Thompson-Eidhammer scheme is DeMott2010 scheme. Although the DeMott2015 scheme has been implemented in the code of the Thompson-Eidhammer scheme, it cannot be used without modifying the code. We modified the code to call the DeMott2015 scheme in Thompson-Eidhammer scheme for the condensation and immersion freezing in our simulations.

We explained the DeMott schemes in detail because we made some modifications in this part of code and would conduct sensitivity experiments on the parameter $c_f$ in the DeMott2015 scheme. To avoid making it too redundant, we did not explain the other schemes in the manuscript. We think that the readers could refer to the references for the other schemes.

Comment #9: Just to be sure, the simulated data plotted in Fig. 3, 4, 5 belong to the test run DUST (not to CTRL), is it correct? Please, specify it.

Response: Yes. We have specified it in the beginning of subsection 5.1.

Comment #10: In Fig. 5, the authors focus their attention on the circled area with dust sources. However, there is an evident difference between the simulated and the observed AOD values over India and South China. Could they explain why?

Response: The high AOD values over India and South China are attributed to anthropogenic aerosols. As we did not include emissions other than dust in the simulations, the high values over these regions could not be produced. Only those high values due to dust aerosol in the circled area were produced. We have clarified it in section 5.1.3.

Minor comments:
Technical Comments / Suggestions:

Lines 30-33: I would stop the first sentence after "scheme" at line 31, and I would join "Results suggest that..." with the last sentence. Here, I would only mention that the ice nucleation scheme is not much sensitive to the calibration factor without specifying the numbers (I think this information is too specific for the Abstract).

Response: Revised as suggested.

Lines 61-62: Please, introduce here the abbreviations CCN and IN and use them in the rest of the text.

Response: Revised.

Line 69: Add reference after "WRF-Chem" or rephrase the sentence, otherwise it seems that GOCART has been implemented by the authors.

Response: The reference has been added.

Lines 74-77: I would change the sentence to: "IN 2014, the aerosol-aware Thompson microphysics scheme, which takes into account the aerosols serving as CLOUD CONDENSATION NUCLEI AND ice nuclei, has been implemented into WRF, enabling the model to explicitly predict the number concentration for cloud droplets AND ICE CRYSTALS (Thompson and Eidhammer, 2014)." Would it be correct?

Response: Revised as suggested.

Lines 82-85: This paragraph should be developed and made clearer.
The first point is the implementation, the second point is the validation plus investigation. Thus, it would be better to split the sentence in two parts and to mention also the validation of the model.
Please, explain what is the meaning of "online simulations" or write differently (probably it is better to rephrase the sentence as the expression "online simulations" does not appear again in the text, so it is not important to introduce it here).
Moreover, I would specify "THE GOCART aerosol model" at line 82: "we AIM to fully couple the aerosol-aware Thompson microphysics scheme with THE GOCART aerosol model in ...".
I think it would be clearer since only the GOCART aerosol mode has been mentioned until this point and only later, at line 95, the reader will discover that GOCART is one of three schemes.

Response: The paragraph has been rewritten.
We have replaced "online simulation" with "WRF-Chem integrates various emission schemes and aerosol mechanisms for simulating the emission, transport, mixing, and chemical transformation of aerosols simultaneously with the meteorology".
Revised.

Line 101: Is there a reference about the implementation of GOCART into WRF-Chem to add here?

Response: The reference has been added.

Lines 104-105: It is not clear the correspondence between the emission schemes and the list of references.
Please, write: \Shao's dust emission scheme (REF) is one of the three dust emission schemes in the GOCART aerosol model. THE OTHER TWO SCHEMES ARE DEFINED BY REF-1

AND REF-2".

Otherwise, simply write: "Shao's dust emission scheme (REF) is one of the dust emission schemes in the GOCART aerosol model" without saying "three".

Response: Revised as suggested.

Lines 113-116: As far as I understood, the aerosol-aware version of the Thompson scheme is the evolution of the Thompson scheme. Please, add the reference for the Thompson scheme at line 113 and for the aerosol-aware Thompson scheme at line 116.

It would be better to write "... and therefore it explicitly predicts the number concentrations of cloud condensation nuclei and ice nuclei as well as the number concentration of cloud droplets and ice crystals ". Or, did I misunderstand the meaning?

Response: Revised as suggested.

Lines 149-152: Nicer to read: "... can be approximated through the mean effective radius (rdust, UNIT) and density (ρdust, UNIT) OF DUST PARTICLES for that size bin:" deleting line 152.

Please, specify always the UNIT, e.g.: (A, UNIT). Also at line 159.

Response: Revised as suggested. The units have been added.

Line 153: More correct should be "DUST number concentration (N, #/kg) for ..." instead of "The aerosol number concentration N ...". Also at line 156.

Response: Revised.

Line 169: According to the reference DeMott et al. 2010, nice;Tk in equation (4) is actually nIN;Tk defined as number concentration of IN (instead of ice crystal number concentration). I would follow the notation and the variable descriptions of the reference.

The same is valid for equation (5).

Then, the authors could add that nIN = nice as generally IN are not enough to deplete supersaturation (if this were the case, the more e_cient IN would nucleate _rst and nice would be less than nIN).

Response: Revised as suggested.

Lines 173, 177: Again, according to DeMott et al., I think it should be "... or low concentration of IN compared ..." and "... or higher concentrations of IN based ...", instead of "ice crystals".

Response: Revised.

Lines 174, 181: If the authors refer to Fig. S1 of DeMott2010, the relationship is between IN number concentrations and aerosol particle number concentrations (not ice crystals).

Similarly at line 181.

Response: Revised.

Lines 201-209: These two paragraphs describe how to implement the GOCART-Thompson scheme, but they are badly written (the explanations are not clear and some of them are redundant).

The authors should check this sub-subsection and rewrite it more clearly.

Response: We have rewritten the subsection and deleted the redundant content.

Lines 229-230: Delete \for the following simulation". I mean: \The mass mixing ratio for dust aerosol in a particular size bin n is then updated FOR the next time step $(t + 1)$:"

It would be more correct to use labels for the time in equation (8), e.g.: Ct+1 = Ct-wetscavt.

Response: Revised.

Line 233: The wet removal of dust is proportional to the concentration of what? Dust number concentration? Please, specify it.

Is there a reference to add for the wet deposition scheme in the GOCART aerosol model?

Response: The wet removal of dust is proportional to the concentration of the number concentration of dust particles. We have clarified it in the revised manuscript.

The reference has been added.

Line 245: Information about the model time step could be added.

Response: The time step for the simulation (120s) has been added.

Line 256: It is better to write here that no other aerosol emissions are considered besides dust (what written at line 358), so the reader knows this when the analysis starts.

Does it mean that ice nucleation in CTRL occurs only via homogeneous nucleation? Please, write it.

Response: We have clarified in section 4 that there were no other aerosol emissions being considered in the simulations.

Ice crystals are produced by both homogeneous nucleation and heterogeneous nucleation in CTRL and DUST run. We have clarified it in section 3. In both CTRL and DUST, the newly–implemented GOCART–Thompson microphysics scheme was used for condensation and immersion freezing; the deposition nucleation is determined by the parameterization of Phillips et al. (Phillips et al., 2008), and the freezing of deliquesced aerosols using the hygroscopic aerosol concentration is parameterized following Koop et al. (Koop et al., 2000), with the background aerosol concentration set to be 1/L, which means that in CTRL, the background aerosol concentration is 1/L.

Lines 264-265: Are there some references for the washout method and the volume-averaging method?

Response: The references have been added.

Lines 274-278: This paragraph should be rearranged and PM10 should be de_ned. Moreover, the word "trend" is not used with its statistical meaning (the authors do not compute any trend of dust concentration, rather they check the magnitude and the behaviour of the temporal series). For this reason it would be better to avoid the use of "trend" (also in the next text) and to use, for instance, behaviour, evolution, etc.

Possibility for this paragraph: "The observations of surface concentration of particulate matter with diameter < 10 µm (PM10) measured at ten environmental monitoring stations were used to examine the capability of the model in reproducing dust levels at the ground surface during the simulation period. The ten stations (indicated by blue dots in Figure 1) were located in or surrounding the dust source areas in East Asia: Jinchang, Gansu Province, Yinchuan, Qinghai Province, Shizuishan, Ningxia Province, Baotou, Inner Mongolia, and Yan'an, Shaanxi Province."

At the end the authors could add some characteristics of the measurements: hourly and which unit?

Response: Revised as suggested.

Lines 334-336: It seems to me that the third bin is more often comparable to the second one than to the fourth one, so I would mention here only "fourth and fifth".

Please, quantify "major part" and "minor fraction", as done for Fig. 2b (lines 340-341).

Response: Revised as suggested.

Lines 337-342: Is the number density vertically integrated? Please, specify it.

In Fig. 2, "ice-friendly aerosol" is written in the caption and "aerosol number" in the y-axis, while the text refers only to dust particles. The fact that dust particles are the only aerosols (ice-friendly aerosols) emitted comes out only later (line 358). It would be better to mention this already in the section of Model Configurations (and write "dust aerosol" in Fig. 2).

Response: The lable of y-axis has been revised into "dust particle number".

Line 348: Please, write that the time series of the simulated concentrations are extracted from the nearest grid point to the geographical coordinates of the stations. Is it correct?

Response: The information has been added.

Lines 349-352: This paragraph should be modi_ed. I do not see that \the surface PM10 concentration was overestimated at one station in Jinchang", the general tendency of the model should be considered before the individual events (the sentence in lines 353-354 could be moved here), and Fig, 3g and h should be also mentioned. Thus, the paragraph could become: \Overall, the model shows a good performance in simulating the dust cycle at THE different LOCATIONS, with EVOLUTION and magnitude of the daily mean PM10 concentration well captured at most of the stations. THE MODEL TENDS TO PRODUCE SURFACE PM10 CONCENTRATIONS LOWER THAN THOSE OBSERVED, AS NO OTHER EMISSIONS WERE CONSIDERED IN THE SIMULATIONS. HOWEVER, the dust eventS on MARCH 21 AND April 26 wERE overestimated BY THE MODEL at the LOCATIONS in ... " c, d, g, h, i, j.

Response: Revised as suggested.

Line 360: To compute the correlation, which data are used? Hourly measurements and which simulated concentrations? Please, specify it.

Response: The correlation was calculated from the daily mean observed surface $PM_{10}$ concentration and the corresponding simulated values from DUST, we have clarified it in section 5.1.1.

Lines 369-372: This paragraph should be a bit modified. It would be better to say firstly the general performance of the model (remembering that the underestimation is due to the fact that there are no other emissions apart from dust) and to point out later the overestimations in the two periods.

Remember to avoid the usage of "trend".

The temporal means of simulated and observed AOD could be added.

The same considerations are valid for the next paragraph regarding SACOL.

Response: Revised as suggested.

Lines 377: In this sub-subsection both modeled results and observations are analysed, therefore, the title could be changed from "Satellite-observational AOD" to "AOD spatial distribution".

Response: Revised as suggested.

Line 385: Is it possible to quantify (percentage?) "lower values"?

Response: AOD values over TD observed by MISR are around 50% lower than those by MODIS in both March and April.

Line 388: Actually, with a first look at the plots in Fig. 5 it does not seem that the model well reproduces the evolution from March to April because the AOD looks higher in March than in April (while the observations show the opposite, as written at line 383). I see that the motivations for such sentence are provided in the next lines but, please, make line 388 clearer (and more modest).

"trend" → "evolution".

Response: Revised as suggested.

Lines 391-393: Add some numbers (AOD means over GD and TD?).

Response: The simulated and observation AOD mean over TD and GD have been added in subsection 6.1.3.

Lines 406-412: Add the mean values of cloud ice mixing ratio and ice crystal number concentration averaged over the domain 1 (?) and the simulated period, for DUST and CTRL.

Response: The mean values of cloud ice mixing ratio and ice crystal number concentration averaged over the domain 1 and the simulation period have been added in subsection 6.2.1.

Lines 421: Write the explanation for the strong positive bias over the southern part of the domain in Fig. 7.

Response: During dust season, the outbreak of cold high system over northeast Asia can bring quantitative dust aerosol down to the South China Sea or even further. In such cases, strong northwestlies swept across the entire China, and brought large amount of dust, especially fine particles, from source areas to the south border of the domain. Besides, the water vapor mixing ratio over south China Sea can be over five times as that over north China. Large amount of ice nuclei transported by winds, combining with abundant water vapor, results in a significant enhancement in the formation of ice crystals over the area.

We have explained it in section 6.2.

Lines 437-440: It is still not clear to me why CTRL and DUST show almost the same ice water path.

Response: The IWP measured by MODIS shown in Figure 8 includes not only cloud ice, but also precipitable ice, such as snow and grauple. The inclusion of dust effect in the simulation greatly affects the amount of cloud ice, but hardly influence the amount of precipitable ice, and cloud ice only accounts for less than 1/10 of the total atmospheric IWC, therefore, the increase of cloud IWC induced by dust did not result in a visible difference between the IWP produced from CTRL and DUST shown in Figure 8.

We meant to demonstrate that the model had the skill to simulate the spatial distribution of the total atmospheric IWC by showing Figure 8. But we decided to delete the figure in the revised manuscript, as it makes little sense to the purpose of this manuscript (demonstrating that the inclusion of dust effect improves the simulation of cloud ice).

Lines 451: Difficult to appreciate that the ice water path over dust source regions is higher in DUST than in CTRL. Is it possible to add a temporal-spatial mean computed for this region?

Response: See the response to the above comment. The figure has been deleted in the revised manuscript.

Line 459: Please, specify if the simulated profiles refer to exactly the same time (e.g. 06 UTC, ...) of the observations or if they have been averaged (daily means?).

Response: The simulated profiles are at the same hour with the observations. We have clarified it at the beginning of subsection 6.2.2.

Lines 468-469: The sentence is not clear in my opinion. What do the authors mean with "points"?

Response: The sentence has been revised into "…the CALIPSO observations of IWC are  mostly at the locations where the temperatures is lower than $-40$ °C and the altitude is greater than 6 km poleward to 12 km equatorward…"

Lines 478-479: As both CTRL and DUST use the newly-implemented GOCART-Thompson scheme (as written at line 252), I think this sentence is not technically correct: the higher IWC values are due to the fact that in DUST the effect of dust is considered (and not to the use of the GOCART-Thompson microphysics scheme). Is it correct? Like it is written at lines 530-531.

Response: The sentence has been corrected.

Lines 505-508: According to the same considerations written above, please, rephrase also this sentence.

Response: Corrected.

Line 520: Before analysing the single cases, please, describe the main discrepancies: peaks of DUST are always lower in altitude and in magnitude.

Response: The statement has been added in subsection 6.2.3.

Lines 545-549: Not well formulated.

Moreover, why is the calibration factor linked to the relative humidity? It is not clear to me the reasoning which leads the authors to their conclusion.

Response: The paragraph has been rewritten and moved to the end of subsection 6.3.1.

As ice nucleation occurs only in a super-saturated atmosphere with respect to water vapor, the ice nucleation process would be terminated in the GOCART-Thompson microphysics scheme when the environmental $RH_i$ is lower than the threshold $RH_i$, which was set to 105% for the simulations in this study. The consistency in the simulated IWC with increasing $c_f$ for the cases in Figure 11 indicates that in these cases, the environmental $RH_i$ had already reached below 105% when $c_f$ was set to 3, meaning that the water vapor available for freezing into ice crystals has been consumed up with $c_f$ equal to 3, therefore, increasing $c_f$ could not lead to a further increase in simulated IWC. Given the above, lowering the threshold $RH_i$ might result in an enhancement of the simulated IWC.

Lines 551-554: No profile matches the observations, better to write for instance: "... and WAS CLOSER TO the observed profile when ...".
"... set to 3 AND 6, ..".
Delete the part starting with "although": the difference between 3 and 6 is too small that I would not assert this.

Response: Revised as suggested.

Line 563: I would have said 4 or 5... Also at line 603 in the Conclusions.

Response: Revised.

Line 600: Please, add that the model generally underpredicts the IWC.

Response: Revised.

Text corrections / suggestions:
Line 20: Better "... and THE aerosol-aware ..." ?

Response: Revised.

Line 24: I would switch the order because I think that the typical season is spring, not spring 2012: "... during spring, a typical dust-intensive season, in 2012."

Response: Revised.

Line 26: "increases by"➔"increase UP TO".

Response: Revised.

Line 28: "demonstrated"➔"demonstrateS" (the present tense is used in the Abstract).

Response: Revised.

Lines 38-39: Repetition of the word "major", possibly find a synonym.

Response: Revised.

Line 42: "Dust in the atmospherE ALTERS ... ".
Response: Revised.

Line 43: No comma after "atmosphere".
Response: Revised.

Line 45: No comma after "cloud".
Response: Revised.

Line 46: No comma after "nuclei".
Response: Revised.

Line 49: "To date, many studies ..." (with \m" in lower case).
Response: Revised.

Line 54: Better "... on dust-cloud interactionS ...".
Response: Revised.

Line 60: Better "... considering aerosol-cloud interactionS ...";
"... in regional MODELS, ...";
No comma after "schemes".
Response: Revised.

Line 62: No comma after "treated".
Response: Revised.

Lines 59-63: The sentence is too long in my opinion and not very clear in the second part.
Response: The sentence has been rewritten.

Line 75: Remove "droplet" (it is specified in the next line).
Response: Revised.

Line 80: "... climate modelS, which ...".
Response: Revised.

Line 90: "... IN section 6."
Response: Revised.

Line 94: "... and the interactionS in between ...".

Response: Revised.

Line 99: "... sulfate, MINERAL dust, ...".
Response: Revised.

Line 107: No comma after "2016".
Response: Revised.

Line 118: "... number concentrationS using ...".
Response: Revised.

Lines 118-123: The sentence is too long in my opinion, the authors could separate the liquid part from the ice part.
Write in parenthesis only the year.
Response: The sentence has been rewritten.

Line 123: "... water dropletS is ...".
Response: Revised.

Line 128: Repetition of the word "multiple", possibly find a synonym.
Response: Revised.

Line 133: Better "... aerosol-cloud interactionS ...".
Response: Revised.

Lines 138-140: Maybe not very clear: "modifications" (line 138) of what?
One possibility could be: "To investigate the real-time indirect effects of dust aerosol over East Asia, the GOCART model HAS BEEN COUPLED TO the aerosol-aware Thompson microphysics scheme. TO DO THIS, WRF-Chem version 3.8.1 HAS BEEN MODIFIED IN THREE STEPS: modification of ..." or something similar.
Response: Revised.

Line 145: "... the number concentrationS of aerosols are ...";
"... to evaluatE ...".
Response: Revised.

Line 156: Put "n" in italic style, like at line 153.
Response: Revised.

Line 158: No comma after "study".
Response: Revised.

Lines 164-165: I would move the reference to line 164: "(DeMott et al., 2010, hereafter DeMott2010 scheme)".
Response: Revised as suggested.

Lines 166: Delete "to account for condensation and immersion freezing" it is obvious from two lines before.
Response: Revised.

Line 170: Put "a, b, c, d" in italic style.
Response: Revised.

Line 176: Similarly to before, I would write: "(DeMott et al., 2015, hereafter the DeMott2015 scheme)".
Response: Revised as suggested.

Lines 177-178: Repetition of the word "latest", possibly find a synonym.
Response: Revised.

Line 181: "DeMott2015" (with "M" in upper case).
Response: Revised.

Lines 185-187: The last sentence could be deleted, there is nothing new with respect to the sentence at lines 176-178.
Response: The sentence has been deleted.

Line 188: "The number concentration of ice crystals produced by ..." without the word "that" and the singular form for "concentration" (otherwise, later, it should be: "is" → "are" and "that" → "those").
Response: Revised.

Line 193: Add comma after the first "scheme".
Response: Revised.

Lines 210-212: Move "is calculated" before: "... at grid point (i,j,k) is calculated, I.E. the tendency of ...".
Response: Revised.

Line 222: "... the fraction of dust particle for each size bin ($\phi$, UNIT) can be ...".
Response: The unit has been added.

Line 226: "... the loss of dust mass due to the microphysical processes (wetscav, UNIT) for a particular size bin n is ...".
Response: The unit has been added.

Lines 236-237: In my opinion, it would be nicer to specify and describe the two experiments from the beginning, as the characteristics written in the following lines actually regard both of them and not only "A numerical experiment" as written at the start of the sentence. Therefore, I would move the lines 251-252 near to 236-237, e.g.: \TWO numerical experimentS WERE conducted to examine the performance of the newly-implemented GOCART-Thompson microphysics scheme in simulating the ice nucleation process induced by dust in the atmosphere. One control run (CTRL) was conducted without dust and one test run (DUST) was con- ducted with dust, both using the GOCART-Thompson microphysics scheme.". In this case, the first sentence at line 251 should be removed.

Response: Revised as suggested.

Line 238: Where? "... dust events in 2012 OVER EAST ASIA were ...".

Response: Revised.

Line 239: Remove "for this numerical test".

Response: Revised.

Line 241: The analysis has not started yet, therefore: "further" → "the".

Response: Revised.

Line 246: "simulationS".

Response: Revised.

Lines 247-248: No comma after "km".

Response: Revised.

Line 249: Specify here (TD) and (GD), as used in Fig. 1.

Response: Revised.

Line 256: Add comma after "East Asia".
Is "Shao's dust emission" the subject? If yes, "were" → "was".

Response: Revised.

Line 262: ", the gravitational..." → " ; the gravitational...".

Response: Revised.

Lines 265-266: The sentence about CTRL could be moved at line 256, so it is in contrast to DUST. I.e.: "...used to generate dust emission in the test run DUST. As no dust emission is produced in CTRL,...".

Response: Revised as suggested.

Line 269: Write in parenthesis only the year.

Response: Revised.

Line 284: Remove the sentence with the meaning of AOD. It is not necessary. Otherwise, it would be better to add the meanings also for the other quantities (aerosol extinction and single-scattering albedo).
Response: Revised as suggested.

Lines 294, 321: "observes" → "measures".
Response: Revised.

Line 295: "... spectral BAND centred at ...".
Response: Revised.

Line 301: Earth's changes of what? E.g.:?
Response: The sentence has been rewritten.

Line 303: ", such as deserts," can be removed, because it is said before.
Response: Revised.

Line 307: "... at 550 nm..." (remove "a").
Response: Revised.

Line 333: Please, rephrase the sentence after the comma.
Response: The sentence has been rewritten.

Line 339: Or simply: "... between the TWO time series lies in ...".
Response: Revised.

Lines 345-347: \To evaluate the performance of WRF-Chem in reproducing dust emissionS over East Asia, the simulated surface PM10 concentrationS were compared with THE observations from
THE ten environmental monitoring stations located near dust sources and downwind areas (DESCRIBED IN SUBSECTION 4.1)."
Delete "at the ten stations" at line 347.
Response: Revised as suggested.

Lines 355-357: Delete "of" after "all".
Response: Revised.

Line 378: "The spatial distribution of MONTHLY mean simulated AOD was compared with ...".
Response: Revised.

Lines 379-382: It would be nicer to explain firstly what the circled area indicates and to describe later the AOD values inside the circle, so exchange the order.

Response: Revised as suggested.

Line 386: Remove "for the observations".
Add that the similarity is stronger with MODIS.

Response: Revised.

Line 389: "... the mean OBSERVED AOD was higher in the southern part of the Taklimakan Desert than that in the northern part in March and showed an increase ...".

Response: Revised.

Line 398: Given the content of this subsection and the other sub-subsections, I would personally change the title to something like "Cloud ice over East Asia" (similarly to 5.1 Dust over East Asia).

Response: The title has been changed to "Cloud ice over East Asia".

Lines 400-405: Sentence too long and not well written. The part "as the ice nucleation process is triggered by dust particles at appropriate temperature and relative humidity," can be deleted, it is a repetition. A new sentence could then start as: "Figure 6 shows the overall comparison ...".

Response: The sentence has been rewritten.

Line 405: No new line.

Response: Revised.

Line 416: Remove "spatial pattern of the".

Response: Revised.

Line 427: "The mean simulated ice water ..." → "The simulated ice water path AVERAGED over the simulation period ...". The same at line 593.

Response: Revised.

Line 432: Remove last "in".

Response: Revised.

Line 434: No comma after "Hymalayas".

Response: Revised.

Line 439: Remove "in the simulation,".

Response: Revised.

Line 448: "overestimation" → "underestimation".

Response: Revised.

Lines 456-527: Use sometimes "IWC" (defined at line 323) instead of "ice water content".

Response: Revised.

Line 458: Make reference to Fig. 9 and Fig. 10 (not to Fig. 11 which is described in the next sub-subsection).

Response: Revised.

Line 485: Remove "through areas with heavy dust load", it is not really so.

Response: Revised.

Line 490: It is not really "east coast"...

Response: Revised.

Line 495: "well" → "better".

Response: Revised.

Line 497: "in East China" → "from the dust sources".

Response: Revised.

Line 497: "in to" → "into".

Response: Revised.

Line 514: Fig. 9 and Fig. 10 (not Fig. 11).

Response: Revised.

Lines 515-515: Remove these lines, which belong to the caption of the figure.

Response: Revised.

Line 517: Remove "the simulation for".

Response: Revised.

Line 526: "are" → "were" (the authors have always used the past tense).

Response: Revised.

Line 527: "in" → "with respect to".

Response: Revised.

Line 527: "for" → "of".

Response: Revised.

Lines 537-540: Repetition of the word "from", possibly find a synonym somewhere.

Response: Revised.

Line 539: "THREE OTHER experimentS WERE conducted to investigate the ..." should be clearer.

Response: Revised.

Lines 545-548: Change the word "coagulation".

Response: Revised.

Line 555: "profile" → profileS".

Response: Revised.

Line 558: Remove "at 7 km", it is not needed with "In this case".

Response: Revised.

Line 559: Remove the part after "model", it is a repetition of what written before.

Response: Revised.

Line 571: No comma after "profile".

"peak" → "peakS".

Response: Revised.

Line 573: "... at lower ALTITUDES than the OBSERVED peak...".

Response: Revised.

Lines 573-574: Please, rephrase the sentence after "but".

Response: Revised.

Line 578: "... to couple the GOCART AEROSOL model TO the ...".

Response: Revised.

Line 579: "By applying this NEWLY-IMPLEMENTED microphysics scheme, ...".

Response: Revised.

Lines 580-581: Remove "by the model simultaneously with dust simulation" or explain better.

Response: Revised.

Lines 585-588: "trend" → "evolution".

"... at THE LOCATIONS OF various monitoring stations ...".

Response: Revised.

Line 589: Remove "by serving as ice nuclei".

Response: Revised.

Line 595: "... reproduced by the model over most areas of East Asia, ALTHOUGH SLIGHTLY UN-DERESTIMATED." and then start a new sentence.

Remove "run".

Response: Revised.

Line 598: Remove "and the entire simulation period further", it is not correct because the sentence before refers to the IWC profiles during the dust events, along the satellite orbit or averaged along it (Fig. 9, 10, 11).

Response: Revised.

Line 601: "... calibration factor DEFINED in the DeMott2015 ...".

Response: Revised.

Lines 604: Make it simpler: "... in the model AND IS significantly ...".

Response: Revised.

Figures:

In Fig. 2, Fig. 3, Fig. 4, please, do not write the year 2012 in the tick-labels of the x-axis (to make the plot "lighter"), rather write explicitly the simulation period in the captions.

Response: Revised as suggested.

Fig.1: - try to reduce the size of blue dots or draw the contours in order to show that there are 10 dots;

- in the caption write "Blue dots represent the TEN MONITORING stations used ...";

- in the caption add the meaning of the red triangles.

Response: The size of dots has been reduced, and we also added a zoomed-in map to show the locations of all the stations, with the station name displayed in the map. However, the two stations at Jinchang are too close to each other, so they are still overlapping the other.

Fig.3: Why are there written only 5 different locations (2 per line), while in subsection 4.1 ten locations are listed? It would be clearer to write the ten different locations (one per plot).

Otherwise, if there is a reason to group the stations, it would be better to specify it in subsection 4.1.

Response: The ten stations are located at 5 cities, each city has two stations (with different station code such as XCNAQ77 at Baotou), we grouped the stations by cities. In Figure 3, the station codes as well as the city they are located at are displayed in each figure. We have clarified the reason we grouped the stations in subsection 6.1.1.

Table 1: Same considerations as before (write 10 locations?).

Add the units of the quantities computed in the table.

Response: See the response to the above comment. We have modified Table 1 to display the cities in the first column, and the station code in the second column.

The units have been added.

Fig.5: - is it possible to use the same projections for all plots?

- make country lines a bit thicker;

- near the word AOD (along the color bars), "MODIS", "MISR" and "modeled" could be added (as subscript?);

- the wavelength of MISR should be 555 instead of 550.

Response: We have replotted the plots with the same projection and thicker country lines.

Fig.6: Wrong unit in 6b.

Response: Revised.

Fig.8: - is it possible to use the same projections for all plots?

- make country lines a bit thicker.

Response: We have removed the figure, see the response to minor comment on Lines 437-440.

Fig.9 and 10: - increase the blank space between the two columns of plots;

- write the quantity (IWC) besides the unit (near the color bar);

- what do the red rectangles indicate? They are never used in the text for the explanations, I think they could be removed;

- make country lines a bit thicker;

- It would be nice to plot also the orographic profile (to explain the white areas below the GNIFA values).

Response: The blank space has been added.

IWC has been added.

The red rectangles have been removed.

The country lines have been thickened.

The orographic profiles have been added in the GNIFA profiles.

Fig.11,12,13: - write "DURING the dust events ..";

- "MEAN vertical profiles OF THE observed ice water content from CALIPSO and the simulated ice water content from ...".

Response: Revised.

---

## Referee Report (RR1)

I think that the manuscript has substantially improved at this stage. I have just one comment.
I would ask again the authors to explain more clearly how the heterogeneous nucleation occur in
the CTRL simulation where no aerosol emissions are considered. In the manuscript, the authors
write that "...the freezing of deliquesced aerosols using the hygroscopic aerosol concentration is
parameterized following Koop et al. 2000, with the background aerosol concentration set to be
1/L." at lines 256-257. However, they should explain this concept better in Section 4 so, please,
review the lines 253-258. For example, it would be clearer if they could explain similarly to their
answer to the Editor's comment, i.e.: "By the default setting in the aerosol-aware Thompson
microphysics scheme, heterogeneous nucleation is still activated in the control simulation, but with
a constant ice nuclei concentration of 1 per Liter."
I think that all the other comments I wrote in my previous report have been addressed properly
by the authors. Below, I list only some minor comments.
Assuming a final revision by the authors, I consider the manuscript suitable for the publication.

**Minor comments**

(Line numbers refer to the last version of the manuscript.)

**Line 66:** "is the most abundant aerosols": aerosols $\longrightarrow$ aerosol.

**Lines 70-83:** Thanks for the additional information about WRF and WRF-Chem. I personally find the
order of the information not super clear, I mean: the authors say that the aerosol-aware
Thompson-Eidhammer microphysics scheme has been implemented in WRF, however, in
lines 74-75 they make a contrast mentioning WRF-Chem, which is not introduced yet. It
would be sufficient to write before (e.g. at line 72, after the citation) that the aerosol-aware
Thompson-Eidhammer microphysics scheme has been implemented ALSO in "WRF-Chem,
the Weather Research and Forecast model coupled with Chemistry".

**Line 79:** "or couple" $\longrightarrow$ "or by coupling".

**Lines 90-94:** The paragraph has to be corrected according to the new structure of the manuscript (there
is a new Section now).

**Line 110:** Why is "model" moved from line 111 (after "GOCART aerosol") to line 110 (after "scheme")?
I would leave "GOCART aerosol model".

**Line 134:** "ice crystal" should be changed with "INP".

**Lines 154-156:** In my opinion, the sentence "...WE APPLY the DeMott2015 ice nucleation scheme in the
GOCART-Thompson microphysics scheme TO BE IMPLEMENTED, instead of the De-
Mott2010 scheme, in the default aerosol-aware Thompson-Eidhammer microphysics scheme
to simulate the ice nucleation involving dust." is not as clear as the explanation the authors
wrote in their answers: "Although the DeMott2015 scheme has been implemented in the
code of the Thompson-Eidhammer scheme, it cannot be used without modifying the code.
We modified the code to call the DeMott2015 scheme in Thompson-Eidhammer scheme for
the condensation and immersion freezing in our simulations."

**Line 161:** Please, adjust the sentence "...it is treated as deposition nucleation, and determined by the
parameterization of Phillips et al. (Phillips et al., 2008) is applied to account for deposition
nucleation.". It could be: "...it is treated as deposition nucleation, and determined by the
parameterization of Phillips et al. (2008)" or "...it is treated as deposition nucleation, and the
parameterization of Phillips et al. (2008) is applied to account for deposition nucleation.".
Check how the citation Phillips et al. (Phillips et al., 2008) is written.

**Line 163:** Check how the citation "Koop et al. (Koop et al., 2000)" is written.

**Line 164:** I would have kept "WRF-Chem", instead of "WRF", as WRF-Chem is the model used for the simulations.

**Lines 233-236:** The word "conducted" is repeated 4 times... it does not sound very nice.

**Line 254:** The sentence "The newly-implemented GOCART-Thompson microphysics scheme." lacks a verb. Maybe it has been forgotten.

**Lines 255-257:** Check how the citations "Phillips et al. (Phillips et al., 2008)" and "Koop et al. (Koop et al., 2000)" are written.

**Line 328:** "The number concentrationS of dust particles over East Asia were vertically integrated..."

**Lines 325-327:** In the sentence "The dust particles in the fourth and fifth bins with effective diameters ranging from 6 to 20 $\mu m$ account for around 60% of the total mass of dust aerosols, and dust particles with diameters smaller than 6 $\mu m$ account for around 40% of the total mass of dust aerosols.", it is obvious that the second percentage is 40%. The authors could simply write: "The dust particles in the fourth and fifth bins with effective diameters ranging from 6 to 20 $\mu m$ account for the major part (around 60%) of the total mass of dust aerosols."

**Line 340:** "...the simulated $PM_{10}$ concentrationS were extracted...".

**Line 342:** The sentence "...thus here were five groups in Figure 3." does not sound completely correct.

**Line 344, 348:** "PM10" $\longrightarrow$ "$PM_{10}$".

**Line 345:** Just as reminder to the reader: "...as no other emissions were considered in the simulations APART FROM DUST.".

**Line 346:** I am sorry, I do not see overestimations in Fig. 3e, rather in Fig. 3g and 3h.

**Line 375:** Please, add the reference to Fig. 5 already here: "...from MODIS and MISR products IN FIGURE 5.", otherwise, the high values of AOD are described before any reference to the figure.

**Lines 382-383:** It is not clear if "respectively" refers to March and April. Please, check the sentence.

**Lines 482-485:** There is a repetition. Please, delete "by considering the effects of dust on ice nucleation process," or "by taking in to account the effect of dust in the ice nucleation process".

**Line 519:** "ro" $\longrightarrow$ "to".

**Lines 540-551:** Better "relative humidity threshold" instead of "threshold relative humidity". Also at lines 579 and 580. Also in the abstract at line 30.

**Line 540:** Here, which is the beginning of a new subsection, I would repeat "with respect to ice" (i.e. "...relative humidity threshold with respect to ice to trigger...").

**Lines 569-572:** I think that this paragraph refers to the old Fig. 8, which has been removed.

**Line 579:** "3 or 4" is not consistent with "4 or 5" at line 529 (in my opinion the second option is better).

---

## Author Response (AR2)

Comments:

I have to say that I share a major concern on an issue identified by two of the reviewers about the implementation of the competition between heterogeneous and homogeneous ice nucleation in the cirrus regime. This is described in part 1, but is also crucial for the results in part 2. If the background aerosol concentration for homogeneous freezing of deliquesced aerosols is set to an unrealistically low value of 1/L, all effects by adding dust are drastically exaggerated or can have the wrong sign. Furthermore, I still miss details on whether and how deposition ice nucleation (parameterized by Phillips et al 2008) is coupled to the dust concentration.

Response:

1. We agree that the exclusion of water-friendly aerosols for this process in the simulations is unreasonable and might cause an exaggeration of the effect of dust on the cloud IWP. To address this concern, we have the simulations with actual number concentration of water-friendly aerosols instead of using a background aerosol number concentration of 1/L for Koop's parameterization scheme. In revised manuscript, water-friendly aerosols were included in both CTRL and DUST simulations. The number concentration of water-friendly aerosols was read from the pre-given climatological aerosol profile derived from long-term simulations of global climate models. The aerosol profile was provided by Dr. Thompson and Dr. Eidhammer and used in their paper for evaluating the performance of the Thompson-Eidhammer microphysics scheme (Thompson and Eidhammer, 2014).

The results in the revised manuscript as well as Figures 6-12 have been updated, but our conclusions remain the same according to the simulation results. We have clarified the new configuration in Section 4 (Line 262-267).

2. The Phillips' parameterization scheme was included in the original Thompson-Eidhammer microphysics scheme and coupled with dust concentration, we did not make any modification on this scheme, and therefore we did not elaborate it in our manuscript. When the relative humidity with respect to ice ($RH_i$) is above 105% and relative humidity with respect to water ($RH_w$) is above 98.5%, it is counted as condensation and immersion freezing, and calculated by DeMott2015 scheme; When $RH_i$ is above 105% and $RH_w$ is below 98.5%, it is treated as deposition nucleation, and determined by the Phillips' parameterization scheme (Phillips et al., 2008). We have clarified it in Subsection 2.2 (line 130-135).

We deeply thank the anonymous referee again for the suggestions and comments. All the concerns of the reviewer have been addressed and the responses to each comment are listed below.

General comment:

I would ask again the authors to explain more clearly how the heterogeneous nucleation occur in the CTRL simulation where no aerosol emissions are considered. In the manuscript, the authors write that "...the freezing of deliquesced aerosols using the hygroscopic aerosol concentration is parameterized following Koop et al. 2000, with the background aerosol concentration set to be 1/L." at lines 256-257. However, they should explain this concept better in Section 4 so, please, review the lines 253-258. For example, it would be clearer if they could explain similarly to their answer to the Editor's comment, i.e.: "By the default setting in the aerosol-aware Thompson microphysics scheme, heterogeneous nucleation is still activated in the control simulation, but with a constant ice nuclei concentration of 1 per Liter."

Response: As the Editor and a referee for Part II of this paper pointed out that the freezing of the deliquesced aerosols has a great contribution to cloud IWP, however, we did not include water-friendly aerosols for this process in our simulations, which is unreasonable and might cause an exaggeration of the effect of dust on the cloud IWP. To address this concern, we have the simulations with actual number concentration of water-friendly aerosols instead of using a background aerosol number concentration of 1/L for Koop's parameterization scheme. In revised manuscript, water-friendly aerosols were included in both CTRL and DUST simulations. The number concentration of water-friendly aerosols was read from the pre-given climatological aerosol profile derived from long-term simulations of global climate models. The aerosol profile was provided by Dr. Thompson and Dr. Eidhammer and used in their paper for evaluating the performance of the Thompson-Eidhammer microphysics scheme (Thompson and Eidhammer, 2014).

The results in the revised manuscript as well as Figures 6-12 have been updated, but our conclusions remain the same according to the simulation results. We have clarified the new configuration in Section 4 (Line 262-267).

Minor comments:

Line 66: "is the most abundant aerosols": aerosols → aerosol.

Response: Corrected.

Lines 70-83: Thanks for the additional information about WRF and WRF-Chem. I personally find the order of the information not super clear, I mean: the authors say that the aerosol-aware Thompson-Eidhammer microphysics scheme has been implemented in WRF, however, in lines 74-75 they make a contrast mentioning WRF-Chem, which is not introduced yet. It would be sufficient to write before (e.g. at line 72, after the citation) that the aerosol-aware Thompson-Eidhammer microphysics scheme has been implemented ALSO in "WRF-Chem, the Weather Research and Forecast model coupled with Chemistry".

Response: Revised as suggested.

Line 79: "or couple" → "or by coupling".

Response: Corrected.

Lines 90-94: The paragraph has to be corrected according to the new structure of the manuscript (there is a new Section now).

Response: Corrected.

Line 110: Why is "model" moved from line 111 (after "GOCART aerosol") to line 110 (after "scheme")? I would leave "GOCART aerosol model".

Response: Corrected.

Line 134: "ice crystal" should be changed with "INP".

Response: Revised.

Lines 154-156: In my opinion, the sentence "...WE APPLY the DeMott2015 ice nucleation scheme in the GOCART-Thompson microphysics scheme TO BE IMPLEMENTED, instead of the DeMott2010 scheme, in the default aerosol-aware Thompson-Eidhammer microphysics scheme to simulate the ice nucleation involving dust." is not as clear as the explanation the authors wrote in their answers: "Although the DeMott2015 scheme has been implemented in the code of the Thompson-Eidhammer scheme, it cannot be used without modifying the code. We modified the code to call the DeMott2015 scheme in Thompson-Eidhammer scheme for the condensation and immersion freezing in our simulations."

Response: Revised as suggested.

Line 161: Please, adjust the sentence "...it is treated as deposition nucleation, and determined by the parameterization of Phillips et al. (Phillips et al., 2008) is applied to account for deposition nucleation.". It could be: "...it is treated as deposition nucleation, and determined by the parameterization of Phillips et al. (2008)" or :...it is treated as deposition nucleation, and the parameterization of Phillips et al. (2008) is applied to account for deposition nucleation.".

Check how the citation Phillips et al. (Phillips et al., 2008) is written.

Response: Revised.

Line 163: Check how the citation "Koop et al. (Koop et al., 2000)" is written.

Response: Corrected.

Line 164: I would have kept "WRF-Chem", instead of "WRF", as WRF-Chem is the model used for the simulations.

Response: Revised.

Lines 233-236: The word "conducted" is repeated 4 times... it does not sound very nice.

Response: We have rewritten the sentence.

Line 254: The sentence "The newly-implemented GOCART-Thompson microphysics scheme." lacks a verb. Maybe it has been forgotten.

Response: Revised.

Lines 255-257: Check how the citations "Phillips et al. (Phillips et al., 2008)" and "Koop et al. (Koop et al., 2000)" are written.

Response: Corrected.

Line 328: "The number concentrationS of dust particles over East Asia were vertically integrated..."

Response: Corrected.

Lines 325-327: In the sentence "The dust particles in the fourth and fifth bins with effective diameters ranging from 6 to 20 _m account for around 60% of the total mass of dust aerosols, and dust particles with diameters smaller than 6 _m account for around 40% of the total mass of dust aerosols.", it is obvious that the second percentage is 40%. The authors could simply write: "The dust particles in the fourth and fifth bins with effective diameters ranging from 6 to 20 µm account for the major part (around 60%) of the total mass of dust aerosols."

Response: Revised as suggested.

Line 340: "...the simulated PM10 concentrationS were extracted...".

Response: Corrected.

Line 342: The sentence "...thus here were five groups in Figure 3." does not sound completely correct.
Response: We have rewritten the sentence.

Line 344, 348: "PM10" → PM10".
Response: Corrected.

Line 345: Just as reminder to the reader: "...as no other emissions were considered in the simulations APART FROM DUST.".
Response: Revised as suggested.

Line 346: I am sorry, I do not see overestimations in Fig. 3e, rather in Fig. 3g and 3h.
Response: It was a mistake, we have revised it.

Line 375: Please, add the reference to Fig. 5 already here: "...from MODIS and MISR products IN FIGURE 5.", otherwise, the high values of AOD are described before any reference to the figure.
Response: Revised as suggested.

Lines 382-383: It is not clear if "respectively" refers to March and April. Please, check the sentence.
Response: We have added "March and April" before "respectively".

Lines 482-485: There is a repetition. Please, delete "by considering the effects of dust on ice nucleation process," or "by taking in to account the effect of dust in the ice nucleation process".
Response: Revised.

Line 519: "ro" → "to".
Response: Corrected.

Lines 540-551: Better "relative humidity threshold" instead of "threshold relative humidity". Also at lines 579 and 580. Also in the abstract at line 30.
Line 540: Here, which is the beginning of a new subsection, I would repeat "with respect to ice" (i.e. "...relative humidity threshold with respect to ice to trigger...").
Response: We have replace threshold relative humidity with respect to ice with $RH_i$ in the updated manuscript.

Lines 569-572: I think that this paragraph refers to the old Fig. 8, which has been removed.

Response: Revised.

Line 579: "3 or 4" is not consistent with "4 or 5" at line 529 (in my opinion the second option is better).

Response: Revised.

[revised manuscript text omitted]

---

## Author Response (AR3)

Dear Editor,

According to your suggestion, we have replaced Figure 6 with 2D histograms with color scales representing the number of grid points in specific value bins, and the related context in the manuscript has also been revised.

Again, we are deeply grateful for your handling of our manuscript, as well as your comments and suggestions.

Best regards,

Lin Su

School of Science

The Hong Kong University of Science and Technology